# Double Thompson Sampling for Dueling Bandits

**Huasen Wu**
University of California, Davis
hswu@ucdavis.edu

**Xin Liu**
University of California, Davis
xinliu@ucdavis.edu

## Abstract

In this paper, we propose a Double Thompson Sampling (D-TS) algorithm for dueling bandit problems. As its name suggests, D-TS selects both the first and the second candidates according to Thompson Sampling. Specifically, D-TS maintains a posterior distribution for the preference matrix, and chooses the pair of arms for comparison according to two sets of samples independently drawn from the posterior distribution. This simple algorithm applies to general *Copeland dueling bandits*, including *Condorcet dueling bandits* as a special case. For general Copeland dueling bandits, we show that D-TS achieves $O(K^2 \log T)$ regret. Moreover, using a back substitution argument, we refine the regret to $O(K \log T + K^2 \log \log T)$ in Condorcet dueling bandits and most practical Copeland dueling bandits. In addition, we propose an enhancement of D-TS, referred to as D-TS$^+$, to reduce the regret in practice by carefully breaking ties. Experiments based on both synthetic and real-world data demonstrate that D-TS and D-TS$^+$ significantly improve the overall performance, in terms of regret and robustness.

## 1 Introduction

The dueling bandit problem [1] is a variant of the classical multi-armed bandit (MAB) problem, where the feedback comes in the form of pairwise comparison. This model has attracted much attention as it can be applied in many systems such as information retrieval (IR) [2, 3], where user preferences are easier to obtain and typically more stable. Most earlier work [1, 4, 5] focuses on Condorcet dueling bandits, where there exists an arm, referred to as the Condorcet winner, that beats all other arms. Recent work [6, 7] turns to a more general and practical case of a Copeland winner(s), which is the arm (or arms) that beats the most other arms. Existing algorithms are mainly generalized from traditional MAB algorithms along two lines: 1) UCB (Upper Confidence Bound)-type algorithms, such as RUCB [4] and CCB [6]; and, 2) MED (Minimum Empirical Divergence)-type algorithms, such as RMED [5] and CW-RMED/ECW-RMED [7].

In traditional MAB, an alternative effective solution is Thompson Sampling (TS) [8]. Its principle is to choose the optimal action that maximizes the expected reward according to the randomly drawn belief. TS has been successfully applied in traditional MAB [9, 10, 11, 12] and other online learning problems [13, 14]. In particular, empirical studies in [9] show that TS not only achieves lower regret than other algorithms in practice, but is also more robust as a randomized algorithm.

In the wake of the success of TS in these online learning problems, a natural question is whether and how TS can be applied to dueling bandits to further improve the performance. However, it is challenging to apply the standard TS framework to dueling bandits, because not all comparisons provide information about the system statistics. Specifically, a good learning algorithm for dueling bandits will eventually compare the winner against itself. However, comparing one arm against itself does not provide any statistical information, which is critical in TS to update the posterior distribution. Thus, TS needs to be adjusted so that 1) comparing the winners against themselves is allowed, but, 2) trapping in comparing a non-winner arm against itself is avoided.

In this paper, we propose a Double Thompson Sampling (D-TS) algorithm for dueling bandits, including both Condorcet dueling bandits and general Copeland dueling bandits. As its name suggests, D-TS typically selects both the first and the second candidates according to samples independently drawn from the posterior distribution. D-TS also utilizes the idea of confidence bounds to eliminate the likely non-winner arms, and thus avoids trapping in suboptimal comparisons. Compared to prior studies on dueling bandits, D-TS has both practical and theoretical advantages.

First, the double sampling structure of D-TS better suits the nature of dueling bandits. Launching two independent rounds of sampling provides us the opportunity to select the same arm in both rounds and thus to compare the winners against themselves. This double sampling structure also leads to more extensive utilization of TS (e.g., compared to RCS [3]), and significantly reduces the regret. In addition, this simple framework applies to general Copeland dueling bandits and achieves lower regret than existing algorithms such as CCB [6]. Moreover, as a randomized algorithm, D-TS is more robust in practice.

Second, this double sampling structure enables us to obtain theoretical bounds for the regret of D-TS. As noted in traditional MAB literature [10, 15], theoretical analysis of TS is usually more difficult than UCB-type algorithms. The analysis in dueling bandits is even more challenging because the selection of arms involves more factors and the two selected arms may be correlated. To address this issue, our D-TS algorithm draws the two sets of samples independently. Because their distributions are fully captured by historic comparison results, when the first candidate is fixed, the comparison between it and all other arms is similar to traditional MAB and thus we can borrow ideas from traditional MAB. Using the properties of TS and confidence bounds, we show that D-TS achieves $O(K^2 \log T)$ regret for a general $K$-armed Copeland dueling bandit. More interestingly, the property that the sample distribution only depends on historic comparing results (but not $t$) enables us to refine the regret using a *back substitution* argument, where we show that D-TS achieves $O(K \log T + K^2 \log \log T)$ in Condorcet dueling bandits and many practical Copeland dueling bandits.

Based on the analysis, we further refine the tie-breaking criterion in D-TS and propose its enhancement called D-TS$^+$. D-TS$^+$ achieves the same theoretical regret bound as D-TS, but performs better in practice especially when there are multiple winners.

In summary, the main contributions of this paper are as follows:

- We propose a D-TS algorithm and its enhancement D-TS$^+$ for general Copeland dueling bandits. The double sampling structure suits the nature of dueling bandits and leads to more extensive usage of TS, which significantly reduces the regret.

- We obtain theoretical regret bounds for D-TS and D-TS$^+$. For general Copeland dueling bandits, we show that D-TS and D-TS$^+$ achieve $O(K^2 \log T)$ regret. In Condorcet dueling bandits and most practical Copeland dueling bandits, we further refine the regret bound to $O(K \log T + K^2 \log \log T)$ using a back substitution argument.

- We evaluate the D-TS and D-TS$^+$ algorithms through experiments based on both synthetic and real-world data. The results show that D-TS and D-TS$^+$ significantly improve the overall performance, in terms of regret and robustness, compared to existing algorithms.

## 2 Related Work

Early dueling bandit algorithms study finite-horizon settings, using the "explore-then-exploit" approaches, such as IF [1], BTM [16], and SAVAGE [17]. For infinite horizon settings, recent work has generalized the traditional MAB algorithms to dueling bandits along two lines. First, RUCB [4] and CCB [6] are generalizations of UCB for Condorcet and general Copeland dueling bandits, respectively. In addition, [18] reduces dueling bandits to traditional MAB, which is then solved by UCB-type algorithms, called MutiSBM and Sparring. Second, [5] and [7] extend the MED algorithm to dueling bandits, where they present the lower bound on the regret and propose the corresponding optimal algorithms, including RMED for Condorcet dueling bandits [5], CW-RMED and its computationally efficient version ECW-RMED for general Copeland dueling bandits [7]. Different from such existing work, we study algorithms for dueling bandits from the perspective of TS, which typically achieves lower regret and is more robust in practice.

Dated back to 1933, TS [8] is one of the earliest algorithms for exploration/exploitation tradeoff. Nowadays, it has been applied in many variants of MAB [11, 12, 13] and other more complex problems, e.g., [14], due to its simplicity, good performance, and robustness [9]. Theoretical analysis of TS is much more difficult. Only recently, [10] proposes a logarithmic bound for the standard frequentist expected regret, whose constant factor is further improved in [15]. Moreover [19, 20] derive the bounds for its Bayesian expected regret through information-theoretic analysis.

TS has been preliminarily considered for dueling bandits [3, 21]. In particular, recent work [3] proposes a Relative Confidence Sampling (RCS) algorithm that combines TS with RUCB [4] for Condorcet dueling bandits. Under RCS, the first arm is selected by TS while the second arm is selected according to their RUCB. Empirical studies demonstrate the performance improvement of using RCS in practice, but no theoretical bounds on the regret are provided.

# 3 System Model

We consider a dueling bandit problem with $K$ ($K \geq 2$) arms, denoted by $\mathcal{A} = \{1, 2, \ldots, K\}$. At each time-slot $t > 0$, a pair of arms $(a_t^{(1)}, a_t^{(2)})$ is displayed to a user and a noisy comparison outcome $w_t$ is obtained, where $w_t = 1$ if the user prefers $a_t^{(1)}$ to $a_t^{(2)}$, and $w_t = 2$ otherwise. We assume the user preference is stationary over time and the distribution of comparison outcomes is characterized by the preference matrix $\boldsymbol{P} = [p_{ij}]_{K \times K}$, where $p_{ij}$ is the probability that the user prefers arm $i$ to arm $j$, i.e., $p_{ij} = \mathbb{P}\{i \succ j\}$, $i, j = 1, 2, \ldots, K$. We assume that the displaying order does not affect the preference, and hence, $p_{ij} + p_{ji} = 1$ and $p_{ii} = 1/2$. We say that arm $i$ beats arm $j$ if $p_{ij} > 1/2$.

We study the general Copeland dueling bandits, where the Copeland winner is defined as the arm (or arms) that maximizes the number of other arms it beats [6, 7]. Specifically, the Copeland score is defined as $\sum_{j \neq i} \mathbb{1}(p_{ij} > 1/2)$, and the normalized Copeland score is defined as $\zeta_i = \frac{1}{K-1} \sum_{j \neq i} \mathbb{1}(p_{ij} > 1/2)$, where $\mathbb{1}(\cdot)$ is the indicator function. Let $\zeta^*$ be the highest normalized Copeland score, i.e., $\zeta^* = \max_{1 \leq i \leq K} \zeta_i$. Then the Copeland winner is defined as the arm (or arms) with the highest normalized Copeland score, i.e., $\mathcal{C}^* = \{i : 1 \leq i \leq K, \zeta_i = \zeta^*\}$. Note that the Condorcet winner is a special case of Copeland winner with $\zeta^* = 1$.

A dueling bandit algorithm $\Gamma$ decides which pair of arms to compare depending on the historic observations. Specifically, define a filtration $\mathcal{H}_{t-1}$ as the history before $t$, i.e., $\mathcal{H}_{t-1} = \{a_\tau^{(1)}, a_\tau^{(2)}, w_\tau, \tau = 1, 2, \ldots, t-1\}$. Then a dueling bandit algorithm $\Gamma$ is a function that maps $\mathcal{H}_{t-1}$ to $(a_t^{(1)}, a_t^{(2)})$, i.e., $(a_t^{(1)}, a_t^{(2)}) = \Gamma(\mathcal{H}_{t-1})$. The performance of a dueling bandit algorithm $\Gamma$ is measured by its expected cumulative regret, which is defined as

$$R_\Gamma(T) = \zeta^* T - \frac{1}{2} \sum_{t=1}^{T} \mathbb{E}\big[\zeta_{a_t^{(1)}} + \zeta_{a_t^{(2)}}\big]. \tag{1}$$

The objective of $\Gamma$ is then to minimize $R_\Gamma(T)$. As pointed out in [6], the results can be adapted to other regret definitions because the above definition bounds the number of suboptimal comparisons.

# 4 Double Thompson Sampling

## 4.1 D-TS Algorithm

We present the D-TS algorithm for Copeland dueling bandits, as described in Algorithm 1 (time index $t$ is omitted in pseudo codes for brevity). As its name suggests, the basic idea of D-TS is to select both the first and the second candidates by TS. For each pair $(i, j)$ with $i \neq j$, we assume a beta prior distribution for its preference probability $p_{ij}$. These distributions are updated according to the comparison results $B_{ij}(t-1)$ and $B_{ji}(t-1)$, where $B_{ij}(t-1)$ (resp. $B_{ji}(t-1)$) is the number of time-slots when arm $i$ (resp. $j$) beats arm $j$ (resp. $i$) before $t$. D-TS selects the two candidates by sampling from the posterior distributions.

Specifically, at each time-slot $t$, the D-TS algorithm consists of two phases that select the first and the second candidates, respectively. When choosing the first candidate $a_t^{(1)}$, we first use the RUCB [4] of $p_{ij}$ to eliminate the arms that are unlikely to be the Copeland winner, resulting in a candidate set $\mathcal{C}_t$ (Lines 4 to 6). The algorithm then samples $\theta_{ij}^{(1)}(t)$ from the posterior beta distribution, and the first candidate $a_t^{(1)}$ is chosen by "majority voting", i.e., the arm within $\mathcal{C}_t$ that beats the most arms according to $\theta_{ij}^{(1)}(t)$ will be selected (Lines 7 to 11). The ties are broken randomly here for simplicity and will be refined later in Section 4.3. A similar idea is applied to select the second candidate $a_t^{(2)}$, where new samples $\theta_{ia_t^{(1)}}^{(2)}(t)$ are generated and the arm with the largest $\theta_{ia_t^{(1)}}^{(2)}(t)$ among all arms with $l_{ia_t^{(1)}} \leq 1/2$ is selected as the second candidate (Lines 13 to 14).

The double sampling structure of D-TS is designed based on the nature of dueling bandits, i.e., at each time-slot, two arms are needed for comparison. Unlike RCS [3], D-TS selects both candidates using TS. This leads to more extensive utilization of TS and thus achieves much lower regret. Moreover, the two sets of samples are independently distributed, following the same posterior that is only determined by the comparison statistics $B_{ij}(t-1)$ and $B_{ji}(t-1)$. This property enables us to obtain an $O(K^2 \log T)$ regret bound and further refine it by a back substitution argument, as discussed later.

We also note that RUCB-based elimination (Lines 4 to 6) and RLCB (Relative Lower Confidence Bound)-based elimination (Line 14) are essential in D-TS. Without these eliminations, the algorithm

---

**Algorithm 1** D-TS for Copeland Dueling Bandits

---

1: **Init:** $\boldsymbol{B} \leftarrow \boldsymbol{0}_{K \times K}$; // $B_{ij}$ *is the number of time-slots that the user prefers arm $i$ to $j$.*
2: **for** $t = 1$ **to** $T$ **do**
3:     // *Phase 1: Choose the first candidate* $a^{(1)}$
4:     $\boldsymbol{U} := [u_{ij}]$, $\boldsymbol{L} := [l_{ij}]$, where $u_{ij} = \frac{B_{ij}}{B_{ij} + B_{ji}} + \sqrt{\frac{\alpha \log t}{B_{ij} + B_{ji}}}$, $l_{ij} = \frac{B_{ij}}{B_{ij} + B_{ji}} - \sqrt{\frac{\alpha \log t}{B_{ij} + B_{ji}}}$, if
    $i \neq j$, and $u_{ii} = l_{ii} = 1/2$, $\forall i$; // $\frac{x}{0} := 1$ *for any* $x$.
5:     $\hat{\zeta}_i \leftarrow \frac{1}{K-1} \sum_{j \neq i} \mathbb{1}(u_{ij} > 1/2)$; // *Upper bound of the normalized Copeland score.*
6:     $\mathcal{C} \leftarrow \{i : \hat{\zeta}_i = \max_j \hat{\zeta}_j\}$;
7:     **for** $i, j = 1, \ldots, K$ with $i < j$ **do**
8:         Sample $\theta_{ij}^{(1)} \sim \text{Beta}(B_{ij} + 1, B_{ji} + 1)$;
9:         $\theta_{ji}^{(1)} \leftarrow 1 - \theta_{ij}^{(1)}$;
10:     **end for**
11:     $a^{(1)} \leftarrow \underset{i \in \mathcal{C}}{\arg \max} \sum_{j \neq i} \mathbb{1}(\theta_{ij}^{(1)} > 1/2)$; // *Choosing from $\mathcal{C}$ to eliminate likely non-winner*
    *arms; Ties are broken randomly.*
12:     // *Phase 2: Choose the second candidate* $a^{(2)}$
13:     Sample $\theta_{ia^{(1)}}^{(2)} \sim \text{Beta}(B_{ia^{(1)}} + 1, B_{a^{(1)}i} + 1)$ for all $i \neq a^{(1)}$, and let $\theta_{a^{(1)}a^{(1)}}^{(2)} = 1/2$;
14:     $a^{(2)} \leftarrow \underset{i : l_{ia^{(1)}} \leq 1/2}{\arg \max} \theta_{ia^{(1)}}^{(2)}$; // *Choosing only from uncertain pairs.*
15:     // *Compare and Update*
16:     Compare pair $(a^{(1)}, a^{(2)})$ and observe the result $w$;
17:     Update $\boldsymbol{B}$: $B_{a^{(1)}a^{(2)}} \leftarrow B_{a^{(1)}a^{(2)}} + 1$ if $w = 1$, or $B_{a^{(2)}a^{(1)}} \leftarrow B_{a^{(2)}a^{(1)}} + 1$ if $w = 2$;
18: **end for**

---

may trap in suboptimal comparisons. Consider one extreme case in Condorcet dueling bandits[1]: assume arm 1 is the Condorcet winner with $p_{1j} = 0.501$ for all $j > 1$, and arm 2 is not the Condorcet winner, but with $p_{2j} = 1$ for all $j > 2$. Then for a larger $K$ (e.g., $K > 4$), without RUCB-based elimination, the algorithm may trap in $a_t^{(1)} = 2$ for a long time, because arm 2 is likely to receive higher score than arm 1. This issue can be addressed by RUCB-based elimination as follows: when chosen as the first candidate, arm 2 has a great probability to compare with arm 1; after sufficient comparisons with arm 1, arm 2 will have $u_{21}(t) < 1/2$ with high probability; then arm 2 is likely to be eliminated because arm 1 has $\hat{\zeta}_1(t) = 1 > \hat{\zeta}_2(t)$ with high probability. Similarly, RLCB-based elimination (Line 14, where we restrict to the arms with $l_{ia^{(1)}}(t) \leq 1/2$) is important especially for non-Condorcet dueling bandits. Specifically, $l_{ia_t^{(1)}}(t) > 1/2$ indicates that arm $i$ beats $a_t^{(1)}$ with high probability. Thus, comparing $a_t^{(1)}$ and arm $i$ brings little information gain and thus should be eliminated to minimize the regret.

## 4.2 Regret Analysis

Before conducting the regret analysis, we first introduce certain notations that will be used later.

**Gap to 1/2:** In dueling bandits, an important benchmark for $p_{ij}$ is 1/2, and thus we let $\Delta_{ij}$ be the gap between $p_{ij}$ and 1/2, i.e., $\Delta_{ij} = |p_{ij} - 1/2|$.

**Number of Comparisons:** Under D-TS, $(i, j)$ can be compared in the form of $(a_t^{(1)}, a_t^{(2)}) = (i, j)$ and $(a_t^{(1)}, a_t^{(2)}) = (j, i)$. We consider these two cases separately and define the following counters: $N_{ij}^{(1)}(t) = \sum_{\tau=1}^t \mathbb{1}(a_\tau^{(1)} = i, a_\tau^{(2)} = j)$ and $N_{ij}^{(2)}(t) = \sum_{\tau=1}^t \mathbb{1}(a_\tau^{(1)} = j, a_\tau^{(2)} = i)$. Then the total number of comparisons is $N_{ij}(t) = N_{ij}^{(1)}(t) + N_{ij}^{(2)}(t)$ for $i \neq j$, and $N_{ii}(t) = N_{ii}^{(1)}(t) = N_{ii}^{(2)}(t)$ for $i = j$.

### 4.2.1 $O(K^2 \log T)$ Regret

To obtain theoretical bounds for the regret of D-TS, we make the following assumption:

**Assumption 1:** The preference probability $p_{ij} \neq 1/2$ for any $i \neq j$.

Under Assumption 1, we present the first result for D-TS in general Copeland dueling bandits:

**Proposition 1.** *When applying D-TS with $\alpha > 0.5$ in a Copeland dueling bandit with a preference matrix $P = [p_{ij}]_{K \times K}$ satisfying Assumption 1, its regret is bounded as:*

$$R_{\text{D-TS}}(T) \leq \sum_{i \neq j: p_{ij} < 1/2} \left[ \frac{4\alpha \log T}{\Delta_{ij}^2} + (1 + \epsilon) \frac{\log T}{D(p_{ij} \| 1/2)} \right] + O(\frac{K^2}{\epsilon^2}), \tag{2}$$

*where $\epsilon > 0$ is an arbitrary constant, and $D(p\|q) = p \log \frac{p}{q} + (1 - p) \log \frac{1-p}{1-q}$ is the KL divergence.*

The summation operation in Eq. (2) is conducted over all pairs $(i, j)$ with $p_{ij} < 1/2$. Thus, Proposition 1 states that D-TS achieves $O(K^2 \log T)$ regret in Copeland dueling bandits. To the best of our knowledge, this is the first theoretical bound for TS in dueling bandits. The scaling behavior of this bound with respect to $T$ is order optimal, since a lower bound $\Omega(\log T)$ has been shown in [7]. The refinement of the scaling behavior with respect to $K$ will be discussed later.

Proving Proposition 1 needs to bound the number of comparisons for all pairs $(i, j)$ with $i \notin \mathcal{C}^*$ or $j \notin \mathcal{C}^*$. When fixing the first candidate as $a_t^{(1)} = i$, the selection of the second candidate $a_t^{(2)}$ is similar to a traditional $K$-armed bandit problem with expected utilities $p_{ji}$ ($j = 1, 2, \ldots, K$). However, the analysis is more complex here since different arms are eliminated differently depending on the value of $p_{ji}$. We prove Proposition 1 through Lemmas 1 to 3, which bound the number of comparisons for all suboptimal pairs $(i, j)$ under different scenarios, i.e., $p_{ji} < 1/2$, $p_{ji} > 1/2$, and $p_{ji} = 1/2$ ($j = i \notin \mathcal{C}^*$), respectively.

**Lemma 1.** *Under D-TS, for an arbitrary constant $\epsilon > 0$ and one pair $(i, j)$ with $p_{ji} < 1/2$, we have*

$$\mathbb{E}[N_{ij}^{(1)}(T)] \leq (1 + \epsilon) \frac{\log T}{D(p_{ji} \| 1/2)} + O(\frac{1}{\epsilon^2}). \tag{3}$$

*Proof.* We can prove this lemma by viewing the comparison between the first candidate arm $i$ and its inferiors as a traditional MAB. In fact, it may be even simpler than that in [15] because under D-TS, arm $j$ with $p_{ji} < 1/2$ is competing with arm $i$ with $p_{ii} = 1/2$, which is known and fixed. Then we can bound $\mathbb{E}[N_{ij}^{(1)}(T)]$ using the techniques in [15]. Details can be found in Appendix B.1. $\qquad \square$

**Lemma 2.** *Under D-TS with $\alpha > 0.5$, for one pair $(i, j)$ with $p_{ji} > 1/2$, we have*

$$\mathbb{E}[N_{ij}^{(1)}(T)] \leq \frac{4\alpha \log T}{\Delta_{ji}^2} + O(1). \tag{4}$$

*Proof.* We note that when $a_t^{(1)} = i$, arm $j$ can be selected as $a_t^{(2)}$ only when its RLCB $l_{ji}(t) \leq 1/2$. Then we can bound $\mathbb{E}[N_{ij}^{(1)}(T)]$ by $O(\frac{4\alpha \log T}{\Delta_{ji}^2})$ similarly to the analysis of traditional UCB algorithms [23]. Details can be found in Appendix B.2. $\qquad \square$

**Lemma 3.** *Under D-TS, for any arm $i \notin \mathcal{C}^*$, we have*

$$\mathbb{E}[N_{ii}(T)] \leq O(K) + \sum_{k: p_{ki} > 1/2} \Theta\left( \frac{1}{\Delta_{ki}^2} + \frac{1}{\Delta_{ki}^2 D(1/2 \| p_{ki})} + \frac{1}{\Delta_{ki}^4} \right) = O(K). \tag{5}$$

Before proving Lemma 3, we present an important property for $\hat{\zeta}^*(t) := \max_{1 \leq i \leq K} \hat{\zeta}_i(t)$. Recall that $\zeta^*$ is the maximum normalized Copeland score. Using the concentration property of RUCB (Lemma 6 in Appendix A), the following lemma shows that $\hat{\zeta}^*(t)$ is indeed a UCB of $\zeta^*$.

**Lemma 4.** *For any $\alpha > 0.5$ and $t > 0$, $\mathbb{P}\{\hat{\zeta}^*(t) \geq \zeta^*\} \geq 1 - K \left[ \frac{\log t}{\log(\alpha + 1/2)} + 1 \right] t^{-\frac{2\alpha}{\alpha + 1/2}}$.*

Return to the proof of Lemma 3. To prove Lemma 3, we consider the cases of $\hat{\zeta}^*(t) < \zeta^*$ and $\hat{\zeta}^*(t) \geq \zeta^*$. The former case $\hat{\zeta}^*(t) < \zeta^*$ can be bounded by Lemma 4. For the latter case, we note that when $\hat{\zeta}^*(t) \geq \zeta^*$, the event $(a_t^{(1)}, a_t^{(2)}) = (i, i)$ occurs only if: a) there exists at least one $k \in \mathcal{K}$ with $p_{ki} > 1/2$, such that $l_{ki}(t) \leq 1/2$; and b) $\theta_{ki}^{(2)}(t) \leq 1/2$ for all $k$ with $l_{ki}(t) \leq 1/2$. In this case, we can bound the probability of $(a_t^{(1)}, a_t^{(2)}) = (i, i)$ by that of $(a_t^{(1)}, a_t^{(2)}) = (i, k)$, for $k$ with $p_{ki} > 1/2$ but $l_{ki}(t) \leq 1/2$, where the coefficient decays exponentially. Then we can bound $\mathbb{E}[N_{ii}(T)]$ by $O(1)$ similar to [15]. Details of proof can be found in Appendix B.4.

The conclusion of Proposition 1 then follows by combining Lemmas 1 to 3.

#### 4.2.2 Regret Bound Refinement

In this section, we refine the regret bound for D-TS and reduce its scaling factor with respect to the number of arms $K$.

We sort the arms for each $i \notin \mathcal{C}^*$ in the descending order of $p_{ji}$, and let $(\sigma_{i(1)}, \sigma_{i(2)}, \ldots, \sigma_{i(K)})$ be a permutation of $(1, 2, \ldots, K)$, such that $p_{\sigma_{i(1)}, i} \geq p_{\sigma_{i(2)}, i} \geq \ldots \geq p_{\sigma_{i(K)}, i}$. In addition, for a Copeland winner $i^* \in \mathcal{C}$, let $L_C = \sum_{j=1}^{K} \mathbb{1}(p_{ji^*} > 1/2)$ be the number of arms that beat arm $i^*$. To refine the regret, we introduce an additional no-tie assumption:

**Assumption 2:** For each arm $i \notin \mathcal{C}^*$, $p_{\sigma_{i(L_C+1)}, i} > p_{\sigma_{i(j)}, i}$ for all $j > L_C + 1$.

We present a refined regret bound for D-TS as follows:

**Theorem 1.** *When applying D-TS with $\alpha > 0.5$ in a Copeland dueling bandit with a preference matrix $P = [p_{ij}]_{K \times K}$ satisfying Assumptions 1 and 2, its regret is bounded as:*

$$
R_{\text{D-TS}}(T) \leq \sum_{i \in \mathcal{C}^*} \left[ \sum_{j:p_{ji}>1/2} \frac{4\alpha \log T}{\Delta_{ji}^2} + \sum_{j:p_{ji}<1/2} (1+\epsilon) \frac{\log T}{D(p_{ji}||1/2)} \right] + \sum_{i \notin \mathcal{C}^*} \sum_{j=1}^{L_C+1} \frac{4\alpha \log T}{\Delta_{\sigma_{i(j)}, i}^2}
$$

$$
+ \beta(1+\epsilon)^2 \sum_{i \notin \mathcal{C}^*} \sum_{j=L_C+2}^{K} \frac{\log \log T}{D(p_{\sigma_{i(j)}, i}||p_{\sigma_{i(L_C+1)}, i})} + O(K^3) + O(\frac{K^2}{\epsilon^2}), \tag{6}
$$

*where $\beta > 2$ and $\epsilon > 0$ are constants, and $D(\cdot||\cdot)$ is the KL-divergence.*

In (6), the first term corresponds to the regret when the first candidate $a_t^{(1)}$ is a winner, and is $O(K|\mathcal{C}^*| \log T)$. The second term corresponds to the comparisons between a non-winner arm and its first $L_C + 1$ superiors, which is bounded by $O(K(L_C + 1) \log T)$. The remaining terms correspond to the comparisons between a non-winner arm and the remaining arms, and is bounded by $O(K^2 \log \log T)$. As demonstrated in [6], $L_C$ is relatively small compared to $K$, and can be viewed as a constant. Thus, the total regret $R_{\text{D-TS}}(T)$ is bounded as $R_{\text{D-TS}}(T) = O(K \log T + K^2 \log \log T)$. In particular, this asymptotic trend can be easily seen for Condorcet dueling bandits where $L_C = 0$.

Comparing Eq. (6) with Eq. (2), we can see the difference is the third and fourth terms in (6), which refine the regret of comparing a suboptimal arm and its last $(K - L_C - 1)$ inferiors into $O(\log \log T)$. Thus, to prove Theorem 1, it suffices to show the following additional lemma:

**Lemma 5.** *Under Assumptions 1 and 2, for any suboptimal arm $i \notin \mathcal{C}^*$ and $j > L_C + 1$, we have*

$$
\mathbb{E}[N_{i\sigma_{i(j)}}^{(1)}(T)] \leq \frac{\beta(1+\epsilon)^2 \log \log T}{D(p_{\sigma_{i(j)}, i}||p_{\sigma_{i(L_C+1)}, i})} + O(K) + O(\frac{1}{\epsilon^2}), \tag{7}
$$

*where $\beta > 2$ and $\epsilon > 0$ are constants.*

*Proof.* We prove this lemma using a *back substitution* argument. The intuition is that when fixing the first candidate as $a_t^{(1)} = i$, the comparison between $a_t^{(1)}$ and the other arms is similar to a traditional MAB with expected utilities $p_{ji}$ ($1 \leq j \leq K$). Let $N_i^{(1)}(T) = \sum_{t=1}^{T} \mathbb{1}(a_t^{(1)} = i)$ be the number of time-slots when this type of MAB is played. Using the fact that the distribution of the samples only depends on the historic comparison results (but not $t$), we can show $\mathbb{E}[N_{i,\sigma_{i(j)}}^{(1)}(T)|N_i^{(1)}(T)] = O(\log N_i^{(1)}(T))$, which holds for any $N_i^{(1)}(T)$. We have shown that $\mathbb{E}[N_i^{(1)}(T)] = O(K \log T)$ for any $i \neq \mathcal{C}^*$ when proving Proposition 1. Then, substituting the bound of $\mathbb{E}[N_i^{(1)}(T)]$ back and using the concavity of the $\log(\cdot)$ function, we have $\mathbb{E}[N_{i,\sigma_{i(j)}}^{(1)}(T)] = \mathbb{E}\big[\mathbb{E}[N_{i,\sigma_{i(j)}}^{(1)}(T)|N_i^{(1)}(T)]\big] \leq O(\log \mathbb{E}[N_i^{(1)}(T)]) = O(\log \log T + \log K)$. Details can be found in Appendix C.1 $\qquad\square$

### 4.3 Further Improvement: D-TS$^+$

D-TS is a TS framework for dueling bandits, and its performance can be improved by refining certain components of it. In this section, we propose an enhanced version of D-TS, referred to as D-TS$^+$, that carefully breaks the ties to reduce the regret.

Note that by randomly breaking the ties (Line 11 in Algorithm 1), D-TS tends to explore all potential winners. This may be desirable in certain applications such as restaurant recommendation, where

users may not want to stick to a single winner. However, because of this, the regret of D-TS scales with the number of winners $|\mathcal{C}^*|$ as shown in Theorem 1. To further reduce the regret, we can break the ties according to estimated regret.

Specifically, with samples $\theta_{ij}^{(1)}(t)$, the normalized Copeland score for each arm $i$ can be estimated as $\tilde{\zeta}_i(t) = \frac{1}{K-1} \sum_{j \neq i} \mathbb{1}(\theta_{ij}^{(1)}(t) > 1/2)$. Then the maximum normalized Copeland score is $\tilde{\zeta}^*(t) = \max_i \tilde{\zeta}_i(t)$, and the loss of comparing arm $i$ and arm $j$ is $\tilde{r}_{ij}(t) = \tilde{\zeta}^*(t) - \frac{1}{2}[\tilde{\zeta}_i(t) + \tilde{\zeta}_j(t)]$. For $p_{ij} \neq 1/2$, we need about $\Theta(\frac{\log T}{D(p_{ij}||1/2)})$ time-slots to distinguish it from 1/2 [5]. Thus, when choosing $i$ as the first candidate, the regret of comparing it with all other arms can be estimated by $\tilde{R}_i^{(1)}(t) = \sum_{j:\theta_{ij}^{(1)}(t) \neq 1/2} \tilde{r}_{ij}(t)/D(\theta_{ij}^{(1)}(t)||1/2)$. We propose the following D-TS$^+$ algorithm that breaks the ties to minimize $\tilde{R}_i^{(1)}(t)$.

**D-TS$^+$:** Implement the same operations as D-TS, except for the selection of the first candidate (Line 11 in Algorithm 1) is replaced by the following two steps:

$$\mathcal{A}^{(1)} \leftarrow \{i \in \mathcal{C} : \zeta_i = \max_{i \in \mathcal{C}} \sum_{j \neq i} \mathbb{1}(\theta_{ij}^{(1)} > 1/2)\};$$

$$a^{(1)} \leftarrow \operatorname*{arg\,min}_{i \in \mathcal{A}^{(1)}} \tilde{R}_i^{(1)};$$

D-TS$^+$ only changes the tie-breaking criterion in selecting the first candidate. Thus, the regret bound of D-TS directly applies to D-TS$^+$:

**Corollary 1.** *The regret of D-TS$^+$, $R_{\text{D-TS}^+}(T)$, satisfies inequality* (6) *under Assumptions 1 and 2.*

Corollary 1 provides an upper bound for the regret of D-TS$^+$. In practice, however, D-TS$^+$ performs better than D-TS in the scenarios with multiple winners, as we can see in Section 5 and Appendix D. Our conjecture is that with this regret-minimization criterion, the D-TS$^+$ algorithm tends to focus on one of the winners (if there is no tie in terms of expected regret), and thus reduces the first term in (6) from $O(K|\mathcal{C}^*|\log T)$ to $O(K\log T)$. The proof of this conjecture requires properties for the evolution of the statistics for all arms and the majority voting results based on the Thompson samples, and is complex. This is left as part of our future work.

In the above D-TS$^+$ algorithm, we only consider the regret of choosing $i$ as the first candidate. From Theorem 1, we know that comparing other arms with their superiors will also result in $\Theta(\log T)$ regret. Thus, although the current D-TS$^+$ algorithm performs well in most practical scenarios, one may further improve its performance by taking these additional comparisons into account in $\tilde{R}_i^{(1)}(t)$.

## 5  Experiments

To evaluate the proposed D-TS and D-TS$^+$ algorithms, we run experiments based on synthetic and real-world data. Here we present the results for experiments based on the Microsoft Learning to Rank (MSLR) dataset [24], which provides the relevance for queries and ranked documents. Based on this dataset, [6] derives a preference matrix for 136 rankers, where each ranker is a function that maps a user's query to a document ranking and can be viewed as one arm in dueling bandits. We use the two 5-armed submatrices in [6], one for Condorcet dueling bandit and the other for non-Condorcet dueling bandit. More experiments and discussions can be found in Appendix D [2].

We compare D-TS and D-TS$^+$ with the following algorithms: BTM [16], SAVAGE [17], Sparring [18], RUCB [4], RCS [3], CCB [6], SCB [6], RMED1 [5], and ECW-RMED [7]. For BTM, we set the relaxed factor $\gamma = 1.3$ as [16]. For algorithms using RUCB and RLCB, including D-TS and D-TS$^+$, we set the scale factor $\alpha = 0.51$. For RMED1, we use the same settings as [5], and for ECW-RMED, we use the same setting as [7]. For the "explore-then-exploit" algorithms, BTM and SAVAGE, each point is obtained by resetting the time horizon as the corresponding value. The results are averaged over 500 independent experiments, where in each experiment, the arms are randomly shuffled to prevent algorithms from exploiting special structures of the preference matrix.

In Condorcet dueling bandits, our D-TS and D-TS$^+$ algorithms achieve almost the same performance and both perform much better than existing algorithms, as shown in Fig. 1(a). In particular, compared with RCS, we can see that the full utilization of TS in D-TS and D-TS$^+$ significantly reduces the

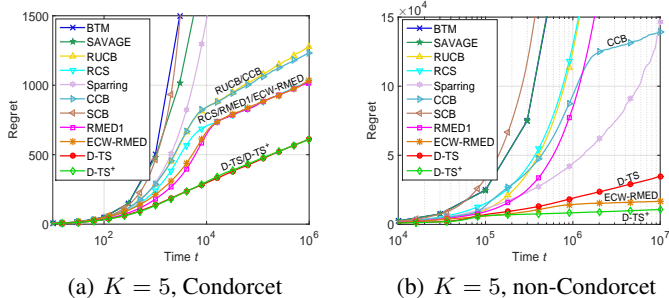
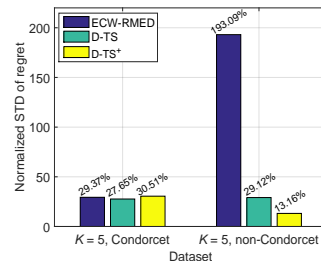

|(a) $K = 5$, Condorcet|(b) $K = 5$, non-Condorcet|
|---|---|

Figure 1: Regret in MSLR dataset. In (b), there are 3 Copeland winners with normalized Copeland score $\zeta^* = 3/4$.

Figure 2: Standard deviation (STD) of regret for $T = 10^6$ (normalized by $R_{\text{ECW}-\text{RMED}}(T)$).

regret. Compared with RMED1 and ECW-RMED, our D-TS and D-TS$^+$ algorithms also perform better. [5] has shown that RMED1 is optimal in Condorcet dueling bandits, not only in the sense of asymptotic order, but also the coefficients in the regret bound. The simulation results show that D-TS and D-TS$^+$ not only achieve the similar slope as RMED1/ECW-RMED, but also converge faster to the asymptotic regime and thus achieve much lower regret. This inspires us to further refine the regret bounds for D-TS and D-TS$^+$ in the future.

In non-Condorcet dueling bandits, as shown in Fig. 1(b), D-TS and D-TS$^+$ significantly reduce the regret compared to the UCB-type algorithm, CCB (e.g., the regret of D-TS$^+$ is less than 10% of that of CCB). Compared with ECW-RMED, D-TS achieves higher regret, mainly because it randomly explores all Copeland winners due to the random tie-breaking rule. With a regret-minimization tie-breaking rule, D-TS$^+$ further reduces the regret, and outperforms ECW-RMED in this dataset. Moreover, as randomized algorithms, D-TS and D-TS$^+$ are more robust to the preference probabilities. As shown in Fig. 2, D-TS and D-TS$^+$ have much smaller regret STD than that of ECW-RMED in the non-Condorcet dataset, where certain preference probabilities (for different arms) are close to 1/2. In particular, the STD of regret for ECW-RMED is almost 200% of its mean value, while it is only 13.16% for D-TS$^+$. In addition, as shown in Appendix D.2.3, D-TS and D-TS$^+$ are also robust to delayed feedback, which is typically batched and provided periodically in practice.

Overall, D-TS and D-TS$^+$ significantly outperform all existing algorithms, with the exception of ECW-RMED. Compared to ECW-RMED, D-TS$^+$ achieves much lower regret in the Condorcet case, lower or comparable regret in the non-Condorcet case, and much more robustness in terms of regret STD and delayed feedback. Thus, the simplicity, good performance, and robustness of D-TS and D-TS$^+$ make them good algorithms in practice.

## 6 Conclusions and Future Work

In this paper, we study TS algorithms for dueling bandits. We propose a D-TS algorithm and its enhanced version D-TS$^+$ for general Copeland dueling bandits, including Condorcet dueling bandits as a special case. Our study reveals desirable properties of D-TS and D-TS$^+$ from both theoretical and practical perspectives. Theoretically, we show that the regret of D-TS and D-TS$^+$ is bounded by $O(K^2 \log T)$ in general Copeland dueling bandits, and can be refined to $O(K \log T + K^2 \log \log T)$ in Condorcet dueling bandits and most practical Copeland dueling bandits. Practically, experimental results demonstrate that these simple algorithms achieve significantly better overall-performance than existing algorithms, i.e., D-TS and D-TS$^+$ typically achieve much lower regret in practice and are robust to many practical factors, such as preference matrix and feedback delay.

Although logarithmic regret bounds have been obtained for D-TS and D-TS$^+$, our analysis relies heavily on the properties of RUCB/RLCB and the regret bounds are likely loose. In fact, we see from experiments that RUCB-based elimination seldom occurs under most practical settings. We will further refine the regret bounds by investigating the properties of TS-based majority-voting. Moreover, results from recent work such as [7] may be leveraged to improve TS algorithms. Last, it is also an interesting future direction to study D-TS type algorithms for dueling bandits with other definition of winners.

**Acknowledgements:** This research was supported in part by NSF Grants CCF-1423542, CNS-1457060, and CNS-1547461. The authors would like to thank Prof. R. Srikant (UIUC), Prof. Shipra Agrawal (Columbia University), Masrour Zoghi (University of Amsterdam), and Dr. Junpei Komiyama (University of Tokyo) for their helpful discussions and suggestions.

## Footnotes

[1]A Borda winner may be more appropriate in this special case [22], and we mainly use it to illustrate the dilemma.

[2]Source codes are available at `https://github.com/HuasenWu/DuelingBandits`.

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
