[Supplementary Material]

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

[3] The optimality of RMED1 is shown under another definition of regret [5], which depends on the probability-gap-to-the-winner (G2W). The trend based on the G2W regret is similar to that in this paper, except for the StrongBorda dataset.

[4]Another reason is that D-TS explores the superiors for each arm sequentially and results in a regret higher than the lower bound in [7].

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

# Appendices

## A Preliminary: Concentration of RUCB/RLCB

We first present the concentration properties of RUCB/RLCB. By relating RUCB/RLCB to UCB/LCB in traditional MAB, we can adjust the results in [23] for RUCB/RLCB as follows.

**Lemma 6.** *1) When $\alpha > 0.5$, for any $(i, j)$ and $t > 0$,*

$$\mathbb{P}\{p_{ij} \geq u_{ij}(t)\} \leq \big[\frac{\log t}{\log(\alpha + 1/2)} + 1\big]t^{-\frac{2\alpha}{\alpha+1/2}}, \tag{8}$$

$$\mathbb{P}\{p_{ij} \leq l_{ij}(t)\} \leq \big[\frac{\log t}{\log(\alpha + 1/2)} + 1\big]t^{-\frac{2\alpha}{\alpha+1/2}}. \tag{9}$$

*2) For any $\alpha > 1/2$,*

$$\sum_{t=1}^{T}\mathbb{P}\{p_{ij} \geq u_{ij}(t)\} \leq \frac{2}{\log(\alpha + 1/2)[2\alpha/(\alpha + 1/2) - 1]^2} = O(1), \tag{10}$$

$$\sum_{t=1}^{T}\mathbb{P}\{p_{ij} \leq l_{ij}(t)\} \leq \frac{2}{\log(\alpha + 1/2)[2\alpha/(\alpha + 1/2) - 1]^2} = O(1). \tag{11}$$

*Proof.* We prove this lemma using the techniques in the proof of Theorem 2.2 in [23].

In fact, RUCB (resp., RLCB) in dueling bandits are essentially the same as UCB (resp., LCB) in traditional MAB. Thus, Part 1) of this lemma can be proved using the peeling argument in [23].

For Part 2), the sum can be bounded by the integration $\int_{1}^{\infty}\big[\frac{\log t}{\log(\alpha+1/2)} + 1\big]t^{-\frac{2\alpha}{\alpha+1/2}}\,\mathrm{d}t$ as in [23]. $\quad\square$

## B Regret Analysis: $O(K^2 \log T)$ Regret

### B.1 Proof of Lemma 1

For a pair $(i, j)$ with $p_{ji} < 1/2$, let $x_{ji}$ be a number satisfying $p_{ji} < x_{ji} < 1/2$. Let $\bar{p}_{ji}(t) = \frac{B_{ji}(t-1)}{B_{ji}(t-1)+B_{ij}(t-1)}$ be the empirical estimation for the probability that arm $j$ beats arm $i$. Define the following events:

$$\mathcal{E}_{ji}^{p}(t) = \{\bar{p}_{ji}(t) < x_{ji}\},$$
$$\mathcal{E}_{ji}^{\theta}(t) = \{\theta_{ji}^{(2)}(t) < 1/2\}.$$

For an event $\mathcal{E}$, we let $\neg\mathcal{E}$ be the event of "not $\mathcal{E}$". Then

$$
\begin{aligned}
\mathbb{E}[N_{ij}^{(1)}(T)] &= \sum_{t=1}^{T}\mathbb{P}\{(a_t^{(1)}, a_t^{(2)}) = (i, j)\} \\
&= \sum_{t=1}^{T}\mathbb{P}\{(a_t^{(1)}, a_t^{(2)}) = (i, j), \mathcal{E}_{ji}^{p}(t), \mathcal{E}_{ji}^{\theta}(t)\} \\
&\quad + \sum_{t=1}^{T}\mathbb{P}\{(a_t^{(1)}, a_t^{(2)}) = (i, j), \mathcal{E}_{ji}^{p}(t), \neg\mathcal{E}_{ji}^{\theta}(t)\} \\
&\quad + \sum_{t=1}^{T}\mathbb{P}\{(a_t^{(1)}, a_t^{(2)}) = (i, j), \neg\mathcal{E}_{ji}^{p}(t)\}.
\end{aligned}
$$

The first term is zero, because $\mathbb{P}\{(a_t^{(1)}, a_t^{(2)}) = (i, j), \mathcal{E}_{ji}^{p}(t), \mathcal{E}_{ji}^{\theta}(t)\} = 0$ for all $t$, due to the fact that $a_t^{(2)} \neq j$ when $\theta_{ji}^{(2)}(t) < 1/2 = \theta_{ii}^{(2)}(t)$.

The second and third terms can be bounded similarly to the analysis of TS in traditional MABs [15]. To see this, we note that when fixing the first candidate as $a_t^{(1)} = i$, the comparison between $i$ and other arms is similar to a traditional MAB problem with expected reward $p_{ji}$ $(1 \leq j \leq K)$. For the case of $p_{ji} < 1/2$, we only need to care about two differences: first, $p_{ii} = 1/2$ is fixed and known; second, in addition to $(a_t^{(1)}, a_t^{(2)}) = (i, j)$, arm $i$ and arm $j$ could also be compared when $(a_t^{(1)}, a_t^{(2)}) = (j, i)$. By capturing the second difference with $N_{ij}(t-1) = N_{ij}^{(1)}(t-1) + N_{ij}^{(2)}(t-1)$, we can leverage the techniques in [15] to prove our results.

Specifically, the second term can be bounded by using the concentration property of the Thompson samples. Letting $L_{ji}(T) = \frac{\log T}{D(x_{ji} || 1/2)}$, similar to the proof of Lemma 4 in [15], we have

$$\sum_{t=1}^{T} \mathbb{P}\{(a_t^{(1)}, a_t^{(2)}) = (i, j), \mathcal{E}_{ji}^p(t), \neg\mathcal{E}_{ji}^\theta(t)\}$$

$$= \sum_{t=1}^{T} \mathbb{P}\{(a_t^{(1)}, a_t^{(2)}) = (i, j), \mathcal{E}_{ji}^p(t), \neg\mathcal{E}_{ji}^\theta(t), N_{ij}(t-1) \leq L_{ji}(T)\}$$

$$+ \sum_{t=1}^{T} \mathbb{P}\{(a_t^{(1)}, a_t^{(2)}) = (i, j), \mathcal{E}_{ji}^p(t), \neg\mathcal{E}_{ji}^\theta(t), N_{ij}(t-1) > L_{ji}(T)\}$$

$$\leq L_{ji}(T) + \sum_{t=1}^{T} \frac{1}{T}$$

$$= \frac{\log T}{D(x_{ji} || 1/2)} + 1. \tag{12}$$

The third term can be bounded similarly to Lemma 3 in [15]. Specifically, let $\tau_n$ be the slot index when $i$ and $j$ are compared for the $n$-th time, including both cases $(a_t^{(1)}, a_t^{(2)}) = (i, j)$ and $(a_t^{(1)}, a_t^{(2)}) = (j, i)$. Let $\tau_0 = 0$. Then, $\bar{p}_{ji}(t)$ is fixed between $\tau_n + 1$ and $\tau_{n+1}$, and $\sum_{t=\tau_n+1}^{\tau_{n+1}} \mathbb{1}((a_t^{(1)}, a_t^{(2)}) = (i, j)) \leq 1$ (it is 0 if the $(n+1)$-th comparison is implemented in the form of $(a_t^{(1)}, a_t^{(2)}) = (j, i)$). Then

$$\sum_{t=1}^{T} \mathbb{P}\{(a_t^{(1)}, a_t^{(2)}) = (i, j), \neg\mathcal{E}_{ji}^p(t)\}$$

$$\leq \sum_{n=0}^{T-1} \mathbb{E}\left[\sum_{t=\tau_n+1}^{\tau_{n+1}} \mathbb{1}((a_t^{(1)}, a_t^{(2)}) = (i, j)) \cdot \mathbb{1}(\neg\mathcal{E}_{ji}^p(t))\right]$$

$$\leq \sum_{n=0}^{T-1} \mathbb{E}\left[\mathbb{1}(\neg\mathcal{E}_{ji}^p(\tau_n+1)) \sum_{t=\tau_n+1}^{\tau_{n+1}} \mathbb{1}((a_t^{(1)}, a_t^{(2)}) = (i, j))\right]$$

$$\leq \sum_{n=0}^{T-1} \mathbb{P}(\neg\mathcal{E}_{ji}^p(\tau_n+1))$$

$$\leq 1 + \sum_{n=1}^{T-1} e^{-nD(x_{ji} || p_{ji})}$$

$$\leq 1 + \frac{1}{D(x_{ji} || p_{ji})}. \tag{13}$$

For any $\epsilon \in (0, 1]$, we choose $x_{ji} \in (p_{ji}, 1/2)$ such that $D(x_{ji} || 1/2) = D(p_{ji} || 1/2)/(1+\epsilon)$, which also implies $\frac{1}{D(x_{ji} || p_{ji})} = O(\frac{1}{\epsilon^2})$ as shown in [15]. The conclusion then follows by combining the bounds for all the above three terms.

## B.2 Proof of Lemma 2

We prove Lemma 2 by using the concentration property of RLCB $l_{ji}(t)$. According to the definition of $N_{ij}^{(1)}(T)$, we have

$$
\begin{aligned}
\mathbb{E}[N_{ij}^{(1)}(T)] &= \sum_{t=1}^{T} \mathbb{P}\{(a_t^{(1)}, a_t^{(2)}) = (i,j)\} \\
&= \sum_{t=1}^{T} \mathbb{P}\{(a_t^{(1)}, a_t^{(2)}) = (i,j), N_{ij}(t-1) \geq \frac{4\alpha \log T}{\Delta_{ji}^2}\} \\
&\quad + \sum_{t=1}^{T} \mathbb{P}\{(a_t^{(1)}, a_t^{(2)}) = (i,j), N_{ij}(t-1) < \frac{4\alpha \log T}{\Delta_{ji}^2}\}.
\end{aligned}
$$

For the first term, we note that when $a_t^{(1)} = i$, arm $j$ can be selected as $a_t^{(2)}$ only when the $l_{ji}(t) \leq 1/2$. When $N_{ij}(t-1) \geq \frac{4\alpha \log T}{\Delta_{ji}^2}$, we have $\Delta_{ji} \geq 2\sqrt{\frac{\alpha \log t}{N_{ij}(t-1)}}$. Thus, $l_{ji}(t) + \Delta_{ji} \geq u_{ji}(t)$. Because $p_{ji} > 1/2$, its RLCB satisfies

$$
\begin{aligned}
\mathbb{P}\{l_{ji}(t) \leq 1/2, N_{ij}(t-1) \geq \frac{4\alpha \log T}{\Delta_{ji}^2}\} &\leq \mathbb{P}\{l_{ji}(t) \leq p_{ji} - \Delta_{ji}, N_{ij}(t-1) \geq \frac{4\alpha \log T}{\Delta_{ji}^2}\} \\
&\leq \mathbb{P}\{u_{ji}(t) \leq p_{ji}\}.
\end{aligned}
$$

Using Lemma 6, we have

$$
\sum_{t=1}^{T} \mathbb{P}\{(a_t^{(1)}, a_t^{(2)}) = (i,j), N_{ij}(t-1) \geq \frac{4\alpha \log T}{\Delta_{ji}^2}\} \leq \sum_{t=1}^{T} \mathbb{P}\{u_{ji}(t) \leq p_{ji}\} = O(1).
$$

For the second term, we can bound it as follows:

$$
\begin{aligned}
&\sum_{t=1}^{T} \mathbb{P}\{(a_t^{(1)}, a_t^{(2)}) = (i,j), N_{ij}(t-1) < \frac{4\alpha \log T}{\Delta_{ji}^2}\} \\
&= \mathbb{E}\left[\sum_{t=1}^{T} \mathbb{1}((a_t^{(1)}, a_t^{(2)}) = (i,j), N_{ij}(t-1) < \frac{4\alpha \log T}{\Delta_{ji}^2})\right] \leq \frac{4\alpha \log T}{\Delta_{ji}^2}, \quad (14)
\end{aligned}
$$

because $\sum_{t=1}^{T} \mathbb{1}((a_t^{(1)}, a_t^{(2)}) = (i,j), N_{ij}(t-1) < \frac{4\alpha \log T}{\Delta_{ji}^2}) \leq \frac{4\alpha \log T}{\Delta_{ji}^2}$ due to the fact that: when $(a_t^{(1)}, a_t^{(2)}) = (i,j)$ at $t$, $N_{ij}(t-1)$ will be increased by 1, but $\mathbb{1}((a_t^{(1)}, a_t^{(2)}) = (i,j), N_{ij}(t-1) < \frac{4\alpha \log T}{\Delta_{ji}^2}) = 0$ as long as $N_{ij}(t-1) \geq \frac{4\alpha \log T}{\Delta_{ji}^2}$.

The conclusion then follows by combining the bounds for the above two terms.

## B.3 Proof of Lemma 4

Let $i^*$ be the Copeland winner (or any one of them if there are multiple Copeland winners) in the dueling bandit. We prove Lemma 4 by analyzing the RUCB $u_{i^*j}(t)$ at $t$. According to Lemma 6, we have that for any $j \neq i^*$,

$$
\mathbb{P}\{u_{i^*j}(t) < p_{i^*j}\} \leq \left[\frac{\log t}{\log(\alpha + 1/2)} + 1\right] t^{-\frac{2\alpha}{\alpha+1/2}}. \quad (15)
$$

Note that $\zeta^* = \frac{1}{K-1}\sum_{j \neq i} \mathbb{1}(p_{i^*j} > 1/2)$. Let $\mathcal{L}_{i^*} = \{j : 1 \leq j \leq K, p_{i^*j} > 1/2\}$ be the set of arms that lose to $i^*$. Thus,

$$
\hat{\zeta}^*(t) < \zeta^* \Rightarrow \exists j \in \mathcal{L}_{i^*}, \text{ such that } u_{i^*j}(t) < p_{i^*j}. \quad (16)
$$

Consider all elements in $\mathcal{L}_{i^*}$, we have

$$
\begin{aligned}
\mathbb{P}\{\hat{\zeta}^*(t) \geq \zeta^*\} &= 1 - \mathbb{P}\{\hat{\zeta}^*(t) < \zeta^*\} \\
&\geq 1 - \mathbb{P}\{\exists j \in \mathcal{L}_{i^*}, \text{ s.t. } u_{i^*j}(t) < p_{i^*j}\} \\
&\geq 1 - |\mathcal{L}_{i^*}|\big[\frac{\log t}{\log(\alpha + 1/2)} + 1\big]t^{-\frac{2\alpha}{\alpha+1/2}} \\
&\geq 1 - K\big[\frac{\log t}{\log(\alpha + 1/2)} + 1\big]t^{-\frac{2\alpha}{\alpha+1/2}}. \tag{17}
\end{aligned}
$$

## B.4   Proof of Lemma 3

To bound the number of time-slots when we compare one non-winner arm against itself, we need to investigate the necessary conditions for this event.

Specifically, when the upper bound of the Copeland score $\hat{\zeta}^*(t) \geq \zeta^*$, the event $(a_t^{(1)}, a_t^{(2)}) = (i, i)$ for $i \notin \mathcal{C}^*$ occurs only if: a) there exists at least one $k \in \mathcal{K}$ with $p_{ki} > 1/2$, such that $l_{ki}(t) \leq 1/2$; and b) $\theta_{ki}^{(2)}(t) \leq 1/2$ for all $k$ with $l_{ki}(t) \leq 1/2$. Now we bound $\mathbb{E}[N_{ii}(T)]$ by bounding the probability of these two conditions.

For $k$ with $p_{ki} > 1/2$, we define the following probability

$$
q_{ki}(t) = \mathbb{P}\{\theta_{ki}^{(2)}(t) > 1/2|\mathcal{H}_{t-1}\}. \tag{18}
$$

Note that the value of $\hat{\zeta}^*(t)$, $l_{ki}(t)$, and $q_{ki}(t)$ depends on the history, and thus is determined by $\mathcal{H}_{t-1}$. Similar to Lemma 1 in [15], we bound the probability of comparing $i$ against itself (accompanied by $\hat{\zeta}^*(t) \geq \zeta^*$ and $l_{ki}(t) \leq 1/2$, which is different from TS for traditional MABs) by that of comparing $i$ with $k$.

**Lemma 7.** *Given $(i, k)$ with $p_{ki} > 1/2$, we have*

$$
\begin{aligned}
&\mathbb{P}\{(a_t^{(1)}, a_t^{(2)}) = (i, i), \hat{\zeta}^*(t) \geq \zeta^*, l_{ki}(t) \leq 1/2|\mathcal{H}_{t-1}\} \\
&\leq \frac{1 - q_{ki}(t)}{q_{ki}(t)}\mathbb{P}\{(a_t^{(1)}, a_t^{(2)}) = (i, k)|\mathcal{H}_{t-1}\}. \tag{19}
\end{aligned}
$$

*Proof.* First of all, the value of $\hat{\zeta}^*(t)$ and $l_{ki}(t)$ depends on $\mathcal{H}_{t-1}$. Thus, if $\mathcal{H}_{t-1}$ satisfies that $\hat{\zeta}^*(t) < \zeta^*$ or $l_{ki}(t) > 1/2$, then (19) holds because the left hand side of is zero.

Now we consider $\mathcal{H}_{t-1}$ satisfying $\hat{\zeta}^*(t) \geq \zeta^*$ and $l_{ki}(t) \leq 1/2$. For the left hand side, we have

$$
\begin{aligned}
&\mathbb{P}\{(a_t^{(1)}, a_t^{(2)}) = (i, i), \hat{\zeta}^*(t) \geq \zeta^*, l_{ki}(t) \leq 1/2|\mathcal{H}_{t-1}\} \\
&= \mathbb{P}\{(a_t^{(1)}, a_t^{(2)}) = (i, i)|\mathcal{H}_{t-1}\} \\
&\leq \mathbb{P}\{\theta_{k'i}^{(2)}(t) \leq 1/2, \forall k', \text{s.t. } l_{k'i}(t) \leq 1/2|\mathcal{H}_{t-1}\} \\
&= \mathbb{P}\{\theta_{ki}^{(2)}(t) \leq 1/2|\mathcal{H}_{t-1}\} \cdot \mathbb{P}\{\theta_{k'i}^{(2)}(t) \leq 1/2, \forall k' \neq k, \text{s.t. } l_{k'i}(t) \leq 1/2|\mathcal{H}_{t-1}\} \\
&= [1 - q_{ki}(t)]\mathbb{P}\{\theta_{k'i}^{(2)}(t) \leq 1/2, \forall k' \neq k, \text{s.t. } l_{k'i}(t) \leq 1/2|\mathcal{H}_{t-1}\}. \tag{20}
\end{aligned}
$$

For the right hand side, we have

$$
\begin{aligned}
&\mathbb{P}\{(a_t^{(1)}, a_t^{(2)}) = (i, k)|\mathcal{H}_{t-1}\} \\
&\geq \mathbb{P}\{\theta_{ki}^{(2)}(t) > 1/2 \geq \theta_{k'i}^{(2)}(t), \forall k' \neq k, \text{s.t. } l_{k'i}(t) \leq 1/2|\mathcal{H}_{t-1}\} \\
&= \mathbb{P}\{\theta_{ki}^{(2)}(t) > 1/2|\mathcal{H}_{t-1}\} \cdot \mathbb{P}\{\theta_{k'i}^{(2)}(t) \leq 1/2, \forall k' \neq k, \text{s.t. } l_{k'i}(t) \leq 1/2|\mathcal{H}_{t-1}\} \\
&= q_{ki}(t)\mathbb{P}\{\theta_{k'i}^{(2)}(t) \leq 1/2, \forall k' \neq i \text{ or } k, \text{s.t. } l_{k'i}(t) \leq 1/2|\mathcal{H}_{t-1}\}. \tag{21}
\end{aligned}
$$

The conclusion then follows by combining (20) and (21). $\qquad\square$

Now we return to the proof of Lemma 3. We divide the probability of $(a_t^{(1)}, a_t^{(2)}) = (i, i)$ into two terms according to the value of $\hat{\zeta}^*(t)$.

$$
\begin{aligned}
&\mathbb{E}[N_{ii}(T)] \\
=\ &\sum_{t=1}^{T} \mathbb{P}\{(a_t^{(1)}, a_t^{(2)}) = (i, i)\} \\
\leq\ &\sum_{t=1}^{T} \mathbb{P}\{(a_t^{(1)}, a_t^{(2)}) = (i, i), \hat{\zeta}^*(t) \geq \zeta^*, \exists k, p_{ki} > 1/2, l_{ki}(t) \leq 1/2\} + \sum_{t=1}^{T} \mathbb{P}\{\hat{\zeta}^*(t) < \zeta^*\} \\
\leq\ &\sum_{k:p_{ki}>1/2} \sum_{t=1}^{T} \mathbb{P}\{(a_t^{(1)}, a_t^{(2)}) = (i, i), \hat{\zeta}^*(t) \geq \zeta^*, l_{ki}(t) \leq 1/2\} + \sum_{t=1}^{T} K\Big[\frac{\log t}{\log(\alpha + 1/2)} + 1\Big]t^{-\frac{2\alpha}{\alpha+1/2}} \\
\leq\ &\sum_{k:p_{ki}>1/2} \sum_{t=1}^{T} \mathbb{P}\{(a_t^{(1)}, a_t^{(2)}) = (i, i), \hat{\zeta}^*(t) \geq \zeta^*, l_{ki}(t) \leq 1/2\} + O(K). \tag{22}
\end{aligned}
$$

In the above equation, we have already bounded the second term by using Lemmas 4 and 6.

Next, we bound the first term by analyzing the bound for each $k$ with $p_{ki} > 1/2$. Let $\tau_n$ be the time-slot index where $k$ and $i$ are compared for the $n$-th time, including both cases $(a_t^{(1)}, a_t^{(2)}) = (i, k)$ and $(a_t^{(1)}, a_t^{(2)}) = (k, i)$, and let $\tau_0 = 0$. Then by Lemma 7, we have that for each $k$ with $p_{ki} > 1/2$,

$$
\begin{aligned}
&\sum_{t=1}^{T} \mathbb{P}\{(a_t^{(1)}, a_t^{(2)}) = (i, i), \hat{\zeta}^*(t) \geq \zeta^*, l_{ki}(t) \leq 1/2\} \\
=\ &\sum_{t=1}^{T} \mathbb{E}\Big[\mathbb{P}\{(a_t^{(1)}, a_t^{(2)}) = (i, i), \hat{\zeta}^*(t) \geq \zeta^*, l_{ki}(t) \leq 1/2 | \mathcal{H}_{t-1}\}\Big] \\
\leq\ &\sum_{t=1}^{T} \mathbb{E}\Big[\frac{1 - q_{ki}(t)}{q_{ki}(t)} \mathbb{P}\{(a_t^{(1)}, a_t^{(2)}) = (i, k) | \mathcal{H}_{t-1}\}\Big] \\
\leq\ &\sum_{t=1}^{T} \mathbb{E}\Big[\mathbb{E}\Big[\frac{1 - q_{ki}(t)}{q_{ki}(t)} \mathbb{1}\big((a_t^{(1)}, a_t^{(2)}) = (i, k)\big) | \mathcal{H}_{t-1}\Big]\Big] \\
\overset{(a)}{=}\ &\sum_{n=0}^{T-1} \mathbb{E}\Big[\frac{1 - q_{ki}(\tau_n + 1)}{q_{ki}(\tau_n + 1)} \sum_{t=\tau_n+1}^{\tau_{n+1}} \mathbb{1}\big((a_t^{(1)}, a_t^{(2)}) = (i, k)\big)\Big] \\
\leq\ &\sum_{n=0}^{T-1} \mathbb{E}\Big[\frac{1}{q_{ki}(\tau_n + 1)} - 1\Big].
\end{aligned}
$$

The equality (a) follows from the fact that the distribution of $\theta_{ki}^{(2)}(t)$ only changes after $k$ and $i$ are compared. According to Lemma 2 in [15], $\mathbb{E}\big[\frac{1}{q_{ki}(\tau_n+1)}\big]$ is bounded as follows:

$$
\begin{aligned}
&\mathbb{E}\Big[\frac{1}{q_{ki}(\tau_n + 1)}\Big] \\
\leq\ &\begin{cases} 1 + \frac{3}{\Delta_{ki}}, & \text{for } n < \frac{8}{\Delta_{ki}}; \\ 1 + \Theta\big(e^{-n\Delta_{ki}^2/2} + \frac{1}{(n+1)\Delta_{ki}^2} e^{-nD(1/2 || p_{ki})} + \frac{1}{e^{n\Delta_{ki}^2/4}-1}\big), & \text{for } n \geq \frac{8}{\Delta_{ki}}. \end{cases}
\end{aligned}
$$

Thus,

$$
\begin{aligned}
&\sum_{t=1}^{T} \mathbb{P}\{(a_t^{(1)}, a_t^{(2)}) = (i, i), \hat{\zeta}^*(t) \geq \zeta^*, l_{ki}(t) \leq 1/2\} \\
\leq\ &\frac{24}{\Delta_{ki}^2} + \sum_{n=0}^{T-1} \Theta\big(e^{-n\Delta_{ki}^2/2} + \frac{1}{(n+1)\Delta_{ki}^2} e^{-nD(1/2 || p_{ki})} + \frac{1}{e^{n\Delta_{ki}^2/4}-1}\big)
\end{aligned}
$$

$$\leq \quad \frac{24}{\Delta_{ki}^2} + \Theta\Big(\frac{1}{\Delta_{ki}^2} + \frac{1}{\Delta_{ki}^2 D(1/2\|p_{ki})} + \frac{1}{\Delta_{ki}^4} + \frac{1}{\Delta_{ki}^2}\Big)$$

$$= \quad \Theta\Big(\frac{1}{\Delta_{ki}^2} + \frac{1}{\Delta_{ki}^2 D(1/2\|p_{ki})} + \frac{1}{\Delta_{ki}^4}\Big).$$

The conclusion then follows by summing over all $k$ with $p_{ki} > 1/2$.

## C   Regret Bound Refinement

Theorem 1 can be proved by combining Lemmas 1 to 3 and Lemma 5. In this appendix, we present the proof of Lemma 5.

### C.1   Proof of Lemma 5

We first consider the event of $(a_t^{(1)}, a_t^{(2)}) = (i, \sigma_{i(j)})$ with $j > L_C + 1$ in two cases with different values of $\hat{\zeta}^*(t)$. Recall that $\zeta^* = (K - L_C)/(K - 1)$ is the maximum normalized Copeland score. Then,

$$
\begin{aligned}
\mathbb{E}[N_{i\sigma_{i(j)}}^{(1)}(T)] \quad &= \quad \sum_{t=1}^{T} \mathbb{P}\{(a_t^{(1)}, a_t^{(2)}) = (i, \sigma_{i(j)})\} \\
&\leq \quad \sum_{t=1}^{T} \mathbb{P}\{(a_t^{(1)}, a_t^{(2)}) = (i, \sigma_{i(j)}), \hat{\zeta}^*(t) \geq \zeta^*\} + \sum_{t=1}^{T} \mathbb{P}\{\hat{\zeta}^*(t) < \zeta^*\} \\
&\leq \quad \sum_{t=1}^{T} \mathbb{P}\{(a_t^{(1)}, a_t^{(2)}) = (i, \sigma_{i(j)}), \hat{\zeta}^*(t) \geq \zeta^*\} + O(K), \quad\quad (23)
\end{aligned}
$$

where the second term is bounded by $\sum_{t=1}^{T} K\big[\frac{\log t}{\log(\alpha+1/2)} + 1\big] t^{-\frac{2\alpha}{\alpha+1/2}} = O(K)$ according to Lemmas 4 and 6.

To bound the first term in (23), with a slight abuse of notation, we choose two numbers $x_{\sigma_{i(j)},i}$ and $y_{\sigma_{i(j)},i}$ such that $p_{\sigma_{i(j)},i} < x_{\sigma_{i(j)},i} < y_{\sigma_{i(j)},i} < p_{\sigma_{i(L_C+1)},i}$, and define the following events:
$$\mathcal{E}_{\sigma_{i(j)},i}^p(t) = \{\bar{p}_{\sigma_{i(j)},i}(t) < x_{\sigma_{i(j)},i}\},$$
$$\mathcal{E}_{\sigma_{i(j)},i}^\theta(t) = \{\theta_{\sigma_{i(j)},i}^{(2)}(t) < y_{\sigma_{i(j)},i}\},$$
where the existence of $x_{\sigma_{i(j)},i}$ and $y_{\sigma_{i(j)},i}$ is guaranteed under Assumption 2. Then, the first term can be decomposed as

$$
\begin{aligned}
&\sum_{t=1}^{T} \mathbb{P}\{(a_t^{(1)}, a_t^{(2)}) = (i, \sigma_{i(j)}), \hat{\zeta}^*(t) \geq \zeta^*\} \\
&= \quad \sum_{t=1}^{T} \mathbb{P}\{(a_t^{(1)}, a_t^{(2)}) = (i, \sigma_{i(j)}), \hat{\zeta}^*(t) \geq \zeta^*, \mathcal{E}_{\sigma_{i(j)},i}^p(t), \mathcal{E}_{\sigma_{i(j)},i}^\theta(t)\} \\
&\quad + \sum_{t=1}^{T} \mathbb{P}\{(a_t^{(1)}, a_t^{(2)}) = (i, \sigma_{i(j)}), \hat{\zeta}^*(t) \geq \zeta^*, \mathcal{E}_{\sigma_{i(j)},i}^p(t), \neg\mathcal{E}_{\sigma_{i(j)},i}^\theta(t)\} \\
&\quad + \sum_{t=1}^{T} \mathbb{P}\{(a_t^{(1)}, a_t^{(2)}) = (i, \sigma_{i(j)}), \hat{\zeta}^*(t) \geq \zeta^*, \neg\mathcal{E}_{\sigma_{i(j)},i}^p(t)\} \\
&\leq \quad \sum_{t=1}^{T} \mathbb{P}\{(a_t^{(1)}, a_t^{(2)}) = (i, \sigma_{i(j)}), \hat{\zeta}^*(t) \geq \zeta^*, \mathcal{E}_{\sigma_{i(j)},i}^p(t), \mathcal{E}_{\sigma_{i(j)},i}^\theta(t)\} \\
&\quad + \sum_{t=1}^{T} \mathbb{P}\{(a_t^{(1)}, a_t^{(2)}) = (i, \sigma_{i(j)}), \mathcal{E}_{\sigma_{i(j)},i}^p(t), \neg\mathcal{E}_{\sigma_{i(j)},i}^\theta(t)\} \\
&\quad + \sum_{t=1}^{T} \mathbb{P}\{(a_t^{(1)}, a_t^{(2)}) = (i, \sigma_{i(j)}), \neg\mathcal{E}_{\sigma_{i(j)},i}^p(t)\}. \quad\quad (24)
\end{aligned}
$$

Now we bound each term in Eq. (24) respectively.

**a) First term:** $\sum_{t=1}^{T} \mathbb{P}\{(a_t^{(1)}, a_t^{(2)}) = (i, \sigma_{i(j)}), \hat{\zeta}^*(t) \geq \zeta^*, \mathcal{E}_{\sigma_{i(j)},i}^p(t), \mathcal{E}_{\sigma_{i(j)},i}^{\theta}(t)\}$

For the first term, we note that when $\hat{\zeta}^*(t) \geq \zeta^*$, the first candidate $a_t^{(1)}$ could be $i$ only when there exists a $j' \leq L_C + 1$, such that $l_{\sigma_{i(j')},i}(t) \leq 1/2$. Thus,

$$\mathbb{P}\{(a_t^{(1)}, a_t^{(2)}) = (i, \sigma_{i(j)}), \hat{\zeta}^*(t) \geq \zeta^*, \mathcal{E}_{\sigma_{i(j)},i}^p(t), \mathcal{E}_{\sigma_{i(j)},i}^{\theta}(t)\}$$

$$\leq \sum_{j'=1}^{L_C+1} \mathbb{P}\{(a_t^{(1)}, a_t^{(2)}) = (i, \sigma_{i(j)}), \hat{\zeta}^*(t) \geq \zeta^*, l_{\sigma_{i(j')},i}(t) \leq 1/2, \mathcal{E}_{\sigma_{i(j)},i}^p(t), \mathcal{E}_{\sigma_{i(j)},i}^{\theta}(t)\}.$$

For each $j' \leq L_C + 1$, define the following probability:

$$q_{j'j}^{(i)}(t) = \mathbb{P}\{\theta_{\sigma_{i(j')},i}^{(2)}(t) > y_{\sigma_{i(j)},i} | \mathcal{H}_{t-1}\}. \tag{25}$$

Similar to Lemma 7, we can show that

$$\mathbb{P}\{(a_t^{(1)}, a_t^{(2)}) = (i, \sigma_{i(j)}), \hat{\zeta}^*(t) \geq \zeta^*, l_{\sigma_{i(j')},i}(t) \leq 1/2, \mathcal{E}_{\sigma_{i(j)},i}^p(t), \mathcal{E}_{\sigma_{i(j)},i}^{\theta}(t)|\mathcal{H}_{t-1}\}$$

$$\leq \frac{1 - q_{j'j}^{(i)}(t)}{q_{j'j}^{(i)}(t)} \mathbb{P}\{(a_t^{(1)}, a_t^{(2)}) = (i, \sigma_{i(j')}), \mathcal{E}_{\sigma_{i(j)},i}^p(t), \mathcal{E}_{\sigma_{i(j)},i}^{\theta}(t)|\mathcal{H}_{t-1}\},$$

and its summation over $t$ can be bounded as

$$\sum_{t=1}^{T} \mathbb{P}\{(a_t^{(1)}, a_t^{(2)}) = (i, \sigma_{i(j)}), \hat{\zeta}^*(t) \geq \zeta^*, l_{\sigma_{i(j')},i}(t) \leq 1/2, \mathcal{E}_{\sigma_{i(j)},i}^p(t), \mathcal{E}_{\sigma_{i(j)},i}^{\theta}(t)\} = O(1).$$

Considering all $j'$ from 1 to $L_C + 1$, we have

$$\sum_{t=1}^{T} \mathbb{P}\{(a_t^{(1)}, a_t^{(2)}) = (i, \sigma_{i(j)}), \hat{\zeta}^*(t) \geq \zeta^*, \mathcal{E}_{\sigma_{i(j)},i}^p(t), \mathcal{E}_{\sigma_{i(j)},i}^{\theta}(t)\} = O(L_C + 1). \tag{26}$$

**b) Second term:** $\sum_{t=1}^{T} \mathbb{P}\{(a_t^{(1)}, a_t^{(2)}) = (i, \sigma_{i(j)}), \mathcal{E}_{\sigma_{i(j)},i}^p(t), \neg\mathcal{E}_{\sigma_{i(j)},i}^{\theta}(t)\}$

We use the back substitution argument to refine the second term to $O(\log\log T)$. When fixing the first candidate as $a_t^{(1)} = i$, the comparison between $a_t^{(1)}$ and other arms is similar to a traditional MAB. Let $N_i^{(1)}(T) = \sum_{t=1}^{T} \mathbb{1}(a_t^{(1)} = i)$ be the number of time-slots when this type of MAB is played, and let

$$L_{ji}^{\beta}(n) = \frac{\beta \log n}{D(x_{\sigma_{i(j)},i} || y_{\sigma_{i(j)},i})}.$$

Then, considering all possible cases of $N_i^{(1)}(T)$ and $N_{i\sigma_{i(j)}}^{(1)}(t-1)$, we have

$$\sum_{t=1}^{T} \mathbb{P}\{(a_t^{(1)}, a_t^{(2)}) = (i, \sigma_{i(j)}), \mathcal{E}_{\sigma_{i(j)},i}^p(t), \neg\mathcal{E}_{\sigma_{i(j)},i}^{\theta}(t)\}$$

$$\leq \sum_{n=0}^{T} \sum_{t=1}^{T} \mathbb{P}\{(a_t^{(1)}, a_t^{(2)}) = (i, \sigma_{i(j)}), \mathcal{E}_{\sigma_{i(j)},i}^p(t), \neg\mathcal{E}_{\sigma_{i(j)},i}^{\theta}(t), N_{i\sigma_{i(j)}}^{(1)}(t-1) \leq L_{ji}^{\beta}(n), N_i^{(1)}(T) = n\}$$

$$+ \sum_{n=0}^{T} \sum_{t=1}^{T} \mathbb{P}\{(a_t^{(1)}, a_t^{(2)}) = (i, \sigma_{i(j)}), \mathcal{E}_{\sigma_{i(j)},i}^p(t), \neg\mathcal{E}_{\sigma_{i(j)},i}^{\theta}(t), N_{i\sigma_{i(j)}}^{(1)}(t-1) > L_{ji}^{\beta}(n), N_i^{(1)}(T) = n\}.$$

For the first case, note that

$$\sum_{t=1}^{T} \mathbb{P}\{(a_t^{(1)}, a_t^{(2)}) = (i, \sigma_{i(j)}), \mathcal{E}_{\sigma_{i(j)},i}^p(t), \neg\mathcal{E}_{\sigma_{i(j)},i}^{\theta}(t), N_{i\sigma_{i(j)}}^{(1)}(t-1) \leq L_{ji}^{\beta}(n), N_i^{(1)}(T) = n\}$$

$$\leq \sum_{t=1}^{T} \mathbb{P}\{(a_t^{(1)}, a_t^{(2)}) = (i, \sigma_{i(j)}), N_{i\sigma_{i(j)}}^{(1)}(t-1) \leq L_{ji}^{\beta}(n) | N_i^{(1)}(T) = n\} \mathbb{P}\{N_i^{(1)}(T) = n\}$$

$$\leq L_{ji}^{\beta}(n) \mathbb{P}\{N_i^{(1)}(T) = n\}, \tag{27}$$

similar to the analysis for Eq. (14).

Then, we have

$$\sum_{n=0}^{T}\sum_{t=1}^{T}\mathbb{P}\{(a_t^{(1)}, a_t^{(2)}) = (i, \sigma_{i(j)}), \mathcal{E}_{\sigma_{i(j)},i}^{p}(t), \neg\mathcal{E}_{\sigma_{i(j)},i}^{\theta}(t), N_{ij_{ik}}^{(1)}(t-1) \le L_{ji}^{\beta}(n), N_i^{(1)}(T) = n\}$$

$$\le \quad \mathbb{E}[L_{ji}^{\beta}(n)] \le \frac{\beta\log(\mathbb{E}[N_i^{(1)}(T)])}{D(x_{\sigma_{i(j)},i}||y_{\sigma_{i(j)},i})} \le \frac{\beta\big(\log\log T + O(\log K)\big)}{D(x_{\sigma_{i(j)},i}||y_{\sigma_{i(j)},i})},$$

where the last inequality follows from the concavity of the $\log(\cdot)$ function and the fact that $\mathbb{E}[N_i^{(1)}(T)] = O(K\log T)$ as shown when proving Proposition 1.

For the second case, let $\tau_m^{(i)}$ be time-slot where $a_t^{(1)} = i$ for the $m$-th time and $\tau_0^{(i)} = 0$. Then

$$\sum_{t=1}^{T}\mathbb{P}\{(a_t^{(1)}, a_t^{(2)}) = (i, \sigma_{i(j)}), \mathcal{E}_{\sigma_{i(j)},i}^{p}(t), \neg\mathcal{E}_{\sigma_{i(j)},i}^{\theta}(t), N_{i\sigma_{i(j)}}^{(1)}(t-1) > L_{ji}^{\beta}(n), N_i^{(1)}(T) = n\}$$

$$\le \quad \mathbb{E}\left[\sum_{t=1}^{T}\mathbb{1}\left((a_t^{(1)}, a_t^{(2)}) = (i, \sigma_{i(j)}), \mathcal{E}_{\sigma_{i(j)},i}^{p}(t), \neg\mathcal{E}_{\sigma_{i(j)},i}^{\theta}(t), N_{i\sigma_{i(j)}}^{(1)}(t-1) > L_{ji}^{\beta}(n), N_i^{(1)}(T) = n\right)\right]$$

$$\le \quad \sum_{m=0}^{n}\mathbb{E}\left[\sum_{t=\tau_m+1}^{\tau_{m+1}}\mathbb{1}\left((a_t^{(1)}, a_t^{(2)}) = (i, \sigma_{i(j)}), \mathcal{E}_{\sigma_{i(j)},i}^{p}(t), \neg\mathcal{E}_{\sigma_{i(j)},i}^{\theta}(t), N_{i\sigma_{i(j)}}^{(1)}(t-1) > L_{ji}^{\beta}(n)\right)\right]$$

$$\overset{(a)}{\le} \quad \sum_{m=0}^{n}\mathbb{E}\left[\mathbb{1}\left((a_{\tau_{m+1}}^{(1)}, a_{\tau_{m+1}}^{(2)}) = (i, \sigma_{i(j)}), \mathcal{E}_{\sigma_{i(j)},i}^{p}(\tau_{m+1}), \neg\mathcal{E}_{\sigma_{i(j)},i}^{\theta}(\tau_{m+1}), N_{i\sigma_{i(j)}}^{(1)}(\tau_{m+1}-1) > L_{ji}^{\beta}(n)\right)\right]$$

$$\le \quad \sum_{m=0}^{n}\mathbb{P}\{\mathcal{E}_{\sigma_{i(j)},i}^{p}(t), \neg\mathcal{E}_{\sigma_{i(j)},i}^{\theta}(t), N_{i\sigma_{i(j)}}^{(1)}(t-1) > L_{ji}^{\beta}(n)\}$$

$$\overset{(b)}{\le} \quad n\cdot\frac{1}{n^{\beta}} = \frac{1}{n^{\beta-1}}, \tag{28}$$

where $(a)$ holds because $a_t^{(1)} = i$ could only happen at $t = \tau_{m+1}^{(i)}$; $(b)$ is true because, the two sets of samples in D-TS are drawn independently and their distributions only depend on the historic comparison results; thus, given $N_{i\sigma_{i(j)}}^{(1)}(t-1) > L_{ji}^{\alpha}(n)$, the events $\mathcal{E}_{\sigma_{i(j)},i}^{p}(t)$ and $\mathcal{E}_{\sigma_{i(j)},i}^{\theta}(t)$ are independent of $t = \tau_{m+1}^{(i)}$, and the probability can be bounded according to the concentration property of Thompson samples (in the proof of Lemma 3 in [15]):

$$\mathbb{P}\{\mathcal{E}_{\sigma_{i(j)},i}^{p}(t), \neg\mathcal{E}_{\sigma_{i(j)},i}^{\theta}(t), N_{i\sigma_{i(j)}}^{(1)}(t-1) > L_{ji}^{\beta}(n)\} \le e^{-L_{ji}^{\beta}(n)D(x_{\sigma_{i(j)},i}||y_{\sigma_{i(j)},i})} = \frac{1}{n^{\beta}}.$$

Then for $\beta > 2$,

$$\sum_{n=0}^{T}\sum_{t=1}^{T}\mathbb{P}\{(a_t^{(1)}, a_t^{(2)}) = (i, \sigma_{i(j)}), \mathcal{E}_{\sigma_{i(j)},i}^{p}(t), \neg\mathcal{E}_{\sigma_{i(j)},i}^{\theta}(t), N_{i\sigma_{i(j)}}^{(1)}(t-1) > L_{ji}^{\beta}(n), N_i^{(1)}(T) = n\}$$

$$\le \quad \sum_{n=0}^{T}\frac{1}{n^{\beta-1}} = O(1). \tag{29}$$

Combining the above two cases, we can bound the second term of Eq. (24) as:

$$\sum_{t=1}^{T}\mathbb{P}\{(a_t^{(1)}, a_t^{(2)}) = (i, \sigma_{i(j)}), \mathcal{E}_{\sigma_{i(j)},i}^{p}(t), \neg\mathcal{E}_{\sigma_{i(j)},i}^{\theta}(t)\} \le \frac{\beta\big(\log\log T + O(\log K)\big)}{D(x_{\sigma_{i(j)},i}||y_{\sigma_{i(j)},i})} + O(1). \tag{30}$$

**c) Third term:** $\sum_{t=1}^{T}\mathbb{P}\{(a_t^{(1)}, a_t^{(2)}) = (i, \sigma_{i(j)}), \neg\mathcal{E}_{\sigma_{i(j)},i}^{p}(t)\}$

For the third term, we can bound it as

$$\sum_{t=1}^{T}\mathbb{P}\{(a_t^{(1)}, a_t^{(2)}) = (i, \sigma_{i(j)}), \neg\mathcal{E}_{\sigma_{i(j)},i}^p(t)\}$$

$$\leq \sum_{t=1}^{T}\mathbb{P}\{(a_t^{(1)}, a_t^{(2)}) = (i, \sigma_{i(j)}), \neg\mathcal{E}_{\sigma_{i(j)},i}^p(t)\}$$

$$\leq \frac{1}{D(x_{\sigma_{i(j)},i}||p_{\sigma_{i(j)},i})} + 1, \tag{31}$$

where the last inequality follows from Lemma 3 in [15].

Combining the analysis for all above three terms, we can bound Eq. (24) by choosing appropriate $x_{\sigma_{i(j)},i}$ and $y_{\sigma_{i(j)},i}$. Specifically, for any $\epsilon > 0$, similar to [15], we can choose appropriate $x_{\sigma_{i(j)},i}$ and $y_{\sigma_{i(j)},i}$ such that $D(x_{\sigma_{i(j)},i}||y_{\sigma_{i(j)},i}) = D(p_{\sigma_{i(j)},i}||p_{\sigma_{i(L_C+1)},i})/(1+\epsilon)^2$ and $\frac{1}{D(x_{\sigma_{i(j)},i}||p_{\sigma_{i(j)},i})} = O(\frac{1}{\epsilon^2})$. The conclusion of Lemma 5 then follows.

## D  Additional Experimental Results

This appendix presents additional experimental results using both synthetic and real-world data, to further evaluate the performance of the proposed algorithms, in comparison to the state-of-the-art schemes.

Because the simulation complexity is $O(K^2 T)$, we adjust the number of independent experiments to save time, and run 500, 100, and 10 independent experiments for $K < 10$, $10 \leq K \leq 100$, and $K > 100$, respectively. In each experiment, the arms are randomly shuffled to prevent algorithms from exploiting special structures of the preference matrix, except for ECW-RMED in the "Gap" dataset, where we run experiments for both fixed and shuffled arm orders.

### D.1  Datasets

#### D.1.1  Condorcet Dueling Bandits

**Cyclic:** A dataset adopted from [5], where the preference matrix is given by Table 1. In this dataset, Arm 1 is the Condorcet winner with $p_{1j} = 0.6$, and the other arms have a cyclic preference relationship with one arm beating another with high probability. Strong transitivity does not hold in this dataset, and the Condorcet winner is not necessary the best arm when comparing all other arms with a fixed arm, i.e., $p_{1i} < \max_j p_{ji}$ for $i \neq 1$.

**StrongBorda:** A 5-armed dueling bandit with a preference matrix in Table 2. In addition to a Condorcet winner, there is a strong Borda winner, which is not the Condorcet winner, but beats the other arms with high probability. To validate the correctness of algorithms, we still treat this problem as a Condorcet dueling bandit problem and try to find the Condorcet winner, although a Borda winner may be more appropriate in this case [22].

**ArXiv:** A 6-armed dueling bandits with a preference matrix given in Table 3, which is derived by conducting pairwise interleaving experiments [16] based on the search engine of ArXiv.org.

**Sushi:** A 16-armed dueling bandits with a preference matrix derived by [5, 7] from a Sushi preference dataset, where the matrix can be found in the appendix of [7].

Table 1: Cyclic

|   | 1 | 2 | 3 | 4 |
|---|---|---|---|---|
| 1 | 0.5 | 0.6 | 0.6 | 0.6 |
| 2 | 0.4 | 0.5 | 0.9 | 0.1 |
| 3 | 0.4 | 0.1 | 0.5 | 0.9 |
| 4 | 0.4 | 0.9 | 0.1 | 0.5 |

Table 2: StrongBorda

|   | 1 | 2 | 3 | 4 | 5 |
|---|---|---|---|---|---|
| 1 | 0.5 | 0.55 | 0.55 | 0.55 | 0.55 |
| 2 | 0.45 | 0.5 | 0.95 | 0.95 | 0.95 |
| 3 | 0.45 | 0.05 | 0.5 | 0.95 | 0.95 |
| 4 | 0.45 | 0.05 | 0.05 | 0.5 | 0.95 |
| 5 | 0.45 | 0.05 | 0.05 | 0.05 | 0.5 |

Table 3: ArXiv

|   | 1 | 2 | 3 | 4 | 5 | 6 |
|---|---|---|---|---|---|---|
| 1 | 0.50 | 0.55 | 0.55 | 0.54 | 0.61 | 0.61 |
| 2 | 0.45 | 0.50 | 0.55 | 0.55 | 0.58 | 0.60 |
| 3 | 0.45 | 0.45 | 0.50 | 0.54 | 0.51 | 0.56 |
| 4 | 0.46 | 0.45 | 0.46 | 0.50 | 0.54 | 0.50 |
| 5 | 0.39 | 0.42 | 0.49 | 0.46 | 0.50 | 0.51 |
| 6 | 0.39 | 0.40 | 0.44 | 0.50 | 0.49 | 0.50 |

### D.1.2  Non-Condorcet Dueling Bandits

**Non-Condorcet Cyclic:** A 9-armed dueling bandit with a preference matrix given by Table 6. In this dataset, there is a cyclic preference relationship among arms, and the arms can be divided into 3 groups with Copeland scores 6, 4, and 2, respectively. Due to this cyclic symmetry, there are multiple Copeland winners with exactly the same performance.

**Non-Condorcet StrongBorda:** Similar to the Condorcet dueling bandits, we consider this case where there is a Borda winner different from the Copeland winner. In this non-Condorcet setting, we even assume that when comparing the Copeland winner and the Borda winner, the user prefers the Borda winner to the Copeland winner with high probability. Again, this is a extreme case used to the validate the correctness of algorithms.

**Gap:** A dataset adopted from [7], which is a 5-armed dueling bandits with a preference matrix given by Table 5. In this dataset, the ratio between the regret bound for ECW-RMED and the regret bound for the optimal CW-RMED algorithm is very large.

**500-Armed Dueling Bandits:** The 500-armed dueling bandit constructed in [6], where there are three Copeland winners that form a cycle and each has a Copeland score 498, and the other arms have Copeland scores ranging from 0 to 496. We use this dataset to evaluate the scaling behaviors of the algorithms.

Table 4: Gap

|   | 1 | 2 | 3 | 4 | 5 |
|---|------|------|------|------|------|
| 1 | 0.5 | 0.8 | 0.8 | 0.51 | 0.2 |
| 2 | 0.2 | 0.5 | 0.8 | 0.2 | 0.8 |
| 3 | 0.2 | 0.2 | 0.5 | 0.8 | 0.8 |
| 4 | 0.49 | 0.8 | 0.2 | 0.5 | 0.2 |
| 5 | 0.8 | 0.2 | 0.2 | 0.8 | 0.5 |

Table 5: Non-Condorcet StrongBorda

|   | 1 | 2 | 3 | 4 | 5 | 6 |
|---|------|------|------|------|------|------|
| 1 | 0.5 | 0.05 | 0.55 | 0.55 | 0.55 | 0.55 |
| 2 | 0.95 | 0.5 | 0.95 | 0.95 | 0.45 | 0.45 |
| 3 | 0.45 | 0.05 | 0.5 | 0.95 | 0.95 | 0.95 |
| 4 | 0.45 | 0.05 | 0.05 | 0.5 | 0.95 | 0.95 |
| 5 | 0.45 | 0.55 | 0.05 | 0.05 | 0.5 | 0.95 |
| 6 | 0.45 | 0.55 | 0.05 | 0.05 | 0.05 | 0.5 |

Table 6: Non-Condorcet Cyclic

|   | 1 | 2 | 3 | 4 | 5 | 6 | 7 | 8 | 9 |
|---|-----|-----|-----|-----|-----|-----|-----|-----|-----|
| 1 | 0.5 | 0.4 | 0.6 | 0.1 | 0.6 | 0.6 | 0.6 | 0.6 | 0.6 |
| 2 | 0.6 | 0.5 | 0.4 | 0.6 | 0.1 | 0.6 | 0.6 | 0.6 | 0.6 |
| 3 | 0.4 | 0.6 | 0.5 | 0.6 | 0.6 | 0.1 | 0.6 | 0.6 | 0.6 |
| 4 | 0.9 | 0.4 | 0.4 | 0.5 | 0.1 | 0.9 | 0.6 | 0.6 | 0.4 |
| 5 | 0.4 | 0.9 | 0.4 | 0.9 | 0.5 | 0.1 | 0.4 | 0.6 | 0.6 |
| 6 | 0.4 | 0.4 | 0.9 | 0.1 | 0.9 | 0.5 | 0.6 | 0.4 | 0.6 |
| 7 | 0.4 | 0.4 | 0.4 | 0.4 | 0.6 | 0.4 | 0.5 | 0.1 | 0.9 |
| 8 | 0.4 | 0.4 | 0.4 | 0.4 | 0.4 | 0.6 | 0.9 | 0.5 | 0.1 |
| 9 | 0.4 | 0.4 | 0.4 | 0.6 | 0.4 | 0.4 | 0.1 | 0.9 | 0.5 |

### D.1.3  MSLR (Condorcet and non-Condorcet)

For the MSLR dataset, we have evaluated the algorithms in the two 5-arm cases in Section 5, where the preference matrices are given by Tables 7 and 8. In this appendix, we also run experiments for larger scale dueling bandits, $K = 16$ and 32, consisting of arms randomly selected from the 136 rankers (in the *MSLR_Informational_PMat.npz* file on `http://bit.ly/nips15data`, [6]; to see the asymptotic performance within an acceptable $T$, we eliminate the arms with $|p_{ij} - 1/2| < 0.003$). The indices of the chosen rankers and their Copeland scores are presented in Table 9, where the arms are indexed from 1. Note that due to the randomness, our chosen rankers are likely different from those in [5] and [7], even for the same $K$.

Table 7: MSLR ($K = 5$, Condorcet)

|   | 1 | 2 | 3 | 4 | 5 |
|---|-------|-------|-------|-------|-------|
| 1 | 0.500 | 0.535 | 0.613 | 0.757 | 0.765 |
| 2 | 0.465 | 0.500 | 0.580 | 0.727 | 0.738 |
| 3 | 0.387 | 0.420 | 0.500 | 0.659 | 0.669 |
| 4 | 0.243 | 0.273 | 0.341 | 0.500 | 0.510 |
| 5 | 0.235 | 0.262 | 0.331 | 0.490 | 0.500 |

Table 8: MSLR ($K = 5$, non-Condorcet)

|   | 1 | 2 | 3 | 4 | 5 |
|---|-------|-------|-------|-------|-------|
| 1 | 0.500 | 0.484 | 0.519 | 0.529 | 0.518 |
| 2 | 0.516 | 0.500 | 0.481 | 0.530 | 0.539 |
| 3 | 0.481 | 0.519 | 0.500 | 0.504 | 0.512 |
| 4 | 0.471 | 0.470 | 0.496 | 0.500 | 0.503 |
| 5 | 0.482 | 0.461 | 0.488 | 0.497 | 0.500 |

Table 9: MSLR ($K = 16$ and $32$, arms are indexed from 1)

| Subset name | Chosen rankers (Copeland score) | Winners (Copeland score) |
|---|---|---|
| K = 16, Condorcet | 10(0), 22(6), 36(12), 58(9), 59(10), 66(5), 67(3), 68(4), 77(2), 98(1), 109(11), 112(7), 115(15), 116(13), 117(8), 125(14) | 115(15) |
| K = 32, Condorcet | 7(9), 21(26), 24(12), 28(5), 32(13), 35(24), 37(15), 38(19), 43(6), 44(4), 45(10), 48(21), 52(3), 56(28), 61(29), 68(8), 71(27), 73(20), 82(14), 85(25), 87(17), 91(11), 96(1), 99(2), 100(0), 105(7), 112(16), 115(31), 117(18), 121(30), 123(22), 133(23) | 115(31) |
| K = 16, non-Condorcet | 2(6), 21(11), 42(4), 60(14), 67(5), 70(8), 79(3), 90(12), 91(7), 98(0), 99(1), 103(2), 106(14), 109(10), 117(9), 130(14) | 60(14), 106(14), 130(14) |
| K = 32, non-Condorcet | 4(8), 8(4), 12(24), 16(1), 20(0), 24(13), 28(7), 32(14), 36(25), 40(26), 44(6), 48(18), 52(5), 56(28), 60(30), 64(22), 68(10), 72(16), 76(19), 80(21), 84(12), 88(17), 92(9), 96(3), 100(2), 104(11), 108(23), 112(15), 116(28), 120(30), 124(20), 128(29) | 60(30), 120(30) |

## D.2 Performance Comparisons

We first analyze the regret performance of algorithms for Condorcet and non-Condorcet dueling bandits, respectively, and then discuss their robustness with respect to preference matrix and delayed feedback, as well as the impact of RUCB/RLCB elimination.

### D.2.1 Condorcet Dueling Bandits

Fig. 3 shows the cumulative regret of all algorithms in Condorcet dueling bandits. From Figs. 3(a) to 3(f), we can see that D-TS and D-TS$^+$ achieve similar performance, because there is a unique winner in the system and few ties occur when choosing the first candidate. Compared to existing algorithms, D-TS and D-TS$^+$ perform much better, except for the StrongBorda dataset, as discussed later.

Compared with the earlier trial RCS, our D-TS and D-TS$^+$ algorithms lead to more extensive utilization of TS and significantly reduce the regret. Although the RCS algorithm achieves better performance than RUCB and CCB in real-world datasets (Figs. 3(c) to 3(f)) by leveraging TS for selecting the first candidate, the improvement is limited. This is because RCS requires a "100%-pass" when using "majority voting" to select the first candidate, i.e., the arm that beats all the other arms with respect to the samples, and randomly picks one if no such an arm exists. Thus, it may miss many opportunities to choose the Condorcet winner as the first candidate. In contrast, using RUCB-based elimination when choosing the first candidate, D-TS/D-TS$^+$ only need a simple "majority voting" rule. Thus, under D-TS/D-TS$^+$, the Condorcet winner can be chosen as the first candidate even it is beaten by (a few) other arms with respect to the samples. In addition, without requiring "100%-pass", D-TS and D-TS$^+$ directly apply to general Copeland dueling bandits. Moreover, by launching another round of sampling for selecting the second candidate, D-TS and D-TS$^+$ further reduce the regret.

Compared with RMED1 and ECW-RMED, D-TS and D-TS$^+$ still perform better. Note that without knowing the existence of a Condorcet winner, ECW-RMED performs slightly worse than RMED1, while achieving similar asymptotic performance. RMED1 has been shown to achieve the optimal asymptotic performance, in both asymptotic order $O(\log T)$ and *coefficients* [5] [3]. Interestingly, D-TS and D-TS$^+$ not only approach the similar asymptotic performance as RMED1 and ECW-RMED, but also achieve a smaller *constant term*. This is because, at the beginning stage, when the empirical estimates for $p_{ij}$ deviate from the true value, RMED-type algorithms may temporally trap in exploring the non-winner arms and result in a very large constant term. In contrast, by TS, the best arm always has a positive probability to be explored [13], and will be identified as the winner much faster. This property makes TS-type algorithms better for the scenarios where the system statistics are not stationary and slowly varying over time.

For the StrongBorda dataset, Fig. 3(b) shows that D-TS and D-TS$^+$ can escape from the suboptimal comparisons and achieve very good performance compared to existing algorithms, except for (Condorcet-)SAVAGE. Somewhat surprisingly, SAVAGE performs best in this dataset with respect to the Copeland-score-based regret. This is because in this dataset, all arms except for the Condorcet winner (arm 1) are beaten by the Borda winner (arm 2) with a preference probability close to 1. Thus, arms 3 to 5 can be eliminated without causing too much loss by comparing with arm 2. This property in fact leads to a very small lower regret bound according to [7]. As an "explore-then-exploit" algorithm and with the awareness of the existence of a Condorcet winner, SAVAGE can eliminate the suboptimal arms quickly and achieve low regret. ECW-RMED does not perform very well in this dataset as its regret bound is much higher than the optimal lower bound. Note that these results do not conflict with the optimality of RMED1 in Condorcet dueling bandits with respect to the regret

defined in [5]. We can see that SAVAGE approaches the same asymptotic performance to RMED1 if we compare them based probability-gap-to-the-winner regret (for other datasets, the trend and relative relationship is similar for both definitions of regret, and the results are omitted here).

(a) Cyclic

(b) StrongBorda

(c) ArXiv

(d) Sushi

(e) MSLR ($K = 16$, Condorcet)

(f) MSLR ($K = 32$, Condorcet)

Figure 3: Cumulative regret in Condorcet dueling bandits.

### D.2.2  Non-Condorcet Dueling Bandits

Fig. 4 presents the cumulative regret in non-Condorcet dueling bandits. Note that one simulation in the large scale 500-armed dueling bandit takes very long time, and we only compare CCB, SCB, ECW-RMED, and D-TS, for this dataset.

From Fig. 4, we can see that the regret of algorithms dedicated to Condorcet dueling bandits, including BTM, RUCB, RCS, and RMED1, grows rapidly, because these algorithms keep exploring for the

Condorcet winner, which does not exist in these datasets. The Sparring algorithm, whose regret is not theoretically guaranteed, also results in a very large regret in non-Condorcet dueling bandits. The SAVAGE algorithm requires to explore all pairs, and usually leads to large regret.

Compared D-TS and D-TS$^+$ with the UCB-type algorithms for general Copeland dueling bandits, CCB and SCB, we can see that D-TS and D-TS$^+$ perform much better. In particular, as shown in Fig. 4(d) for the 500-armed scenario, D-TS still achieves much lower regret than SCB, the scalable version of CCB for large scale systems.

(a) Non-Condorcet Cyclic

(b) Non-Condorcet StrongBorda

(c) Gap

(d) 500-Armed non-Condorcet Dueling Bandits

(e) MSLR ($K = 16$, non-Condorcet)

(f) MSLR ($K = 32$, non-Condorcet)

Figure 4: Cumulative regret in non-Condorcet dueling bandits.

Compared with the recently developed algorithm ECW-RMED, D-TS and D-TS+ usually converge faster to the asymptotic regime and thus can achieve smaller regret for small to relatively large $T$. In theory, the optimal CW-RMED achieves the best asymptotic performance. ECW-RMED

is its efficient but approximate implementation, with a larger coefficient in terms of asymptotic performance. Therefore, in the asymptotic regime, ECW-RMED could outperform D-TS$^+$ in certain scenarios. In practice, however, we notice that D-TS$^+$ could perform better than ECW-RMED, even for relatively large $T$, e.g., $T = 5 \times 10^6$. This is because ECW-RMED estimates the required number of comparisons based on the empirical preference probability. Then ECW-RMED may temporally trap in suboptimal comparisons at the beginning stage when the empirical preference probability is likely to deviate from its true value. In contrast, D-TS and D-TS$^+$ make decisions in a random manner, and the winner(s) has a positive probability to be explored even when the empirical estimates deviate from the true values. Thus, D-TS and D-TS$^+$ usually converge to the asymptotic regime more quickly than ECW-RMED. Because of this, D-TS and D-TS$^+$ may outperform ECW-RMED even for a relatively large $T$, especially when the number of arms is larger. For example, as shown in Fig. 4(e), ECW-RMED performs worse than D-TS$^+$ when $t \leq 5 \times 10^6$, although it has better asymptotic performance. This situation becomes more serious when the number of arms increases, as we can see from Fig. 4(f).

Our D-TS algorithm performs worse than ECW-RMED when there are multiple winners with similar performance, e.g., for the non-Condorcet Cyclic dataset shown in Fig. 4(a). One main reason is that with a random tie-breaking rule, D-TS randomly explores all potential winners in each individual sample path [4], as shown in Fig. 5. Thus, the regret of D-TS scales with the number of winners $|\mathcal{C}^*|$. By carefully breaking the ties, D-TS$^+$ can reduce the regret in many practical scenarios, as shown in Fig. 1(b), Fig. 4(e), and Fig. 4(f). However, the improvement is limited especially when the winners have very similar or exactly the same performance, as shown in Fig. 4(a). From the perspective of regret optimization, this may be a disadvantage of TS, where based on randomly sampled belief, all winners have a positive probability to be explored [13]. However, from the diversity perspective, this may be desirable in the application scenarios such as restaurant recommendation, where users may not want to stick to a single winner.

Figure 5: Diversity of exploitations: distribution of exploitations over different Copeland winners in one sample path. This is calculated when an arm is compared against itself.

### D.2.3 Robustness

We study the robustness of ECW-RMED, D-TS, and D-TS$^+$, with respect to the preference matrix and delayed feedback.

**Influence of preference matrix:** We have seen from Fig. 2 that, when some preference probabilities for different arms are close to 1/2 (5-armed non-Condorcet MSLR dataset), the regret of ECW-RMED fluctuates significantly and has a very large standard deviation, while D-TS and D-TS$^+$ have much smaller regret deviation.

The robustness of D-TS and D-TS$^+$ to the preference matrix can also be seen from the results in the Gap dataset. Recall that in this dataset, the regret bound of ECW-RMED is much larger than the optimal lower bound. By experiments, we find that the performance of ECW-RMED is significantly affected by the order of arms (and essentially the preference matrix). For example,

when the order of arms is fixed as that in Table 5, ECW-RMED performs very well. Specifically, as shown by the dashed line in Fig. 4(c), the regret of ECW-RMED is very small and its asymptotic behavior is even better than its asymptotic bound (similar results can be found in [7]). However, when the arms are randomly shuffled in each experiment, ECW-RMED achieves much larger regret that is consistent with its asymptotic bound. In contrast, D-TS and D-TS$^+$ do not depend on the order of arms and perform much better than ECW-RMED on average in this dataset.

**Influence of delayed feedback:** In practice, it may be difficult and costly to process each individual comparison result immediately. Typically, feedback will be batched and provided periodically, say every $d$ time-slots. We evaluate the algorithm performance with respect to the feedback delay. As we can see from Fig. 6, as the feedback delay increases, the regret of ECW-RMED increases much faster than D-TS and D-TS$^+$. In particular, even in the non-Condorcet Cyclic dataset with multiple winners (Fig. 6(b)), the regret of ECW-RMED becomes larger than D-TS and D-TS$^+$ when the feedback delay is larger than about 300 time-slots.

(a) MSLR ($K = 5$, Condorcet)          (b) Non-Condorcet Cyclic

Figure 6: Influence of feedback delay: the regret when the feedback is batched and provided every $d$ time-slots ($T = 10^6$).

### D.2.4 Impact of RUCB/RLCB Elimination

In this section, we illustrate the necessity of RUCB/RLCB elimination in D-TS and D-TS$^+$. A pure Thompson Sampling algorithms without the auxiliary RUCB/RLCB elimination step seem to be more elegant and may be more efficient (without the limitation of RUCB/RLCB). However, we notify that the RUCB/RLCB elimination is necessary to guarantee sublinear or logarithmic regret in general settings, especially in non-Condorcet dueling bandits.

Specifically, we consider the following "pure D-TS" algorithm, which is similar to D-TS in Algorithm 1, except that the RUCB/RCLB elimination step is ignored, i.e., the candidates $a^{(1)}$ and $a^{(2)}$ are selected from all arms according to the samples $\theta_{ij}^{(1)}$ and $\theta_{ia^{(1)}}^{(2)}$. As shown in Fig 7, pure D-TS may result in large regret in certain scenarios. For example, in the (Condorcet) StrongBorda dataset (Table 2), the Borda winner (Arm 2) beats all the other arms with high probability except for the Condorcet winner (Arm 1), and hence, it is easier for the Borda winner to get more votes at the beginning and to be chosen as the first candidate. Thus, pure D-TS achieves higher regret in this case as shown in Fig 7(a). In non-Condorcet dueling bandits, without RLCB elimination, pure D-TS could achieve linear regret if a Copeland winner is beaten by a non-winner arm. For example, as shown in Fig. 7(b), the algorithm fails to converge to comparing the Copeland winners against themselves, as the non-winner arm will have higher samples with high probability at the second round. By introducing RLCB elimination, D-TS/D-TS$^+$ can avoid trapping in theses suboptimal comparisons and achieve much better performance.

We also point out without the limitation of RUCB/RLCB, pure D-TS may achieve in certain practical scenarios, e.g., the 5-arm MSLR data sets as shown in Figs 7(c) and 7(d). In fact, the RLCB elimination can be ignored in Condorcet dueling bandits, because none of the other arms can beat the Condorcet winner. It is an interesting direction to identify the conditions under which pure D-TS can perform better and obtain its theoretical performance under these conditions.

(a) Condorcet StrongBorda

(b) Non-Condorcet Cyclic

(c) MSLR ($K = 5$, Condorcet)

(d) MSLR ($K = 5$, non-Condorcet)

Figure 7: Impact of RUCB/RLCB elimination: "pure D-TS" is similar to D-TS, except that the RUCB/RLCB elimination is ignored and the candidates are selected among all arms according to the Thompson samples only.

### D.2.5 Summary of Experimental Results

We summarize the performance evaluation results based on synthetic and real-world data:

- In Condorcet dueling bandits, D-TS and D-TS$^+$ achieve similar performance, and both perform much better than existing algorithms that work for unknown/infinite time-horizon settings. This benefits from the double sampling structure that we proposed for D-TS and D-TS$^+$.

- In non-Condorcet dueling bandits, D-TS and D-TS$^+$ performs much better than UCB-type algorithms, CCB and SCB. Again, this benefits from the double sampling structure of D-TS and D-TS$^+$, assisted by RUCB/RLCB elimination that guarantees sublinear or logarithmic regret in general settings.

- Compared with ECW-RMED, D-TS and D-TS$^+$ achieve much better performance in Condorcet dueling bandits, better or close-to performance in non-Condorcet dueling bandits, especially when $T$ is small to relatively large. Furthermore, D-TS and D-TS$^+$ are also much more robust with respect to the preference matrix and delayed feedback.

In practice, we may not know in advance whether we have a Condorcet and non-Condorcet dueling bandit. We may have in practice a time-varying system and delayed feedback. Overall, good performance, and robustness of D-TS and D-TS$^+$ make them strong candidates in practice.