[Reviews · NeurIPS 2016]

Reviewer 1

Summary

The paper develops Thompson-sampling style algorithms for the interesting and relevant problem of dueling bandits, where the best arm has to be learnt using only pairwise preference feedback. The key idea in the main algorithm is to sample twice, and independently, the unknown model from the posterior distribution over models, and essentially pit the best arms for both models against each other. This has the advantage of promoting exploration in early phases and converging to playing the optimal arm against itself in later phases. The authors derive detailed regret bounds for the algorithm for both the Condorcet winner and the more general Copeland winner ground truth models, and perform numerical comparisons of the proposed algorithm with existing, Upper-Confidence-Bound (UCB) style strategies.

Qualitative Assessment

EDIT: I have read the author response. The paper takes on the challenging task of adapting Thompson sampling-style Bayesian-inspired approaches to the dueling bandit problem, and thus I feel that its contributions are significant. The dueling bandit problem, being a true partial monitoring problem (not all actions give nonzero statistical information), is in a sense harder than standard bandits, and hence requires more care in designing the exploration-exploration tradeoff. Although this is not the first time Thompson sampling has been proposed for the problem (e.g., RCS is an earlier attempt), the smart idea of sampling twice and following it up by a UCB-elimination step is quite novel, and supplemented by fairly competitive analytical guarantees. I believe this would open up paths for more in-depth looks at Thompson sampling approaches to solve classes of dueling bandit problems. Detailed comments and observations are as follows. * Without the Condorcet winner assumption, the algorithm is shown to give a O(K^2logT) regret bound, where as the CCB algorithm of Zoghi et al. gives O(K^2 + KlogT) regret, and SCB gives an even better regret bound of O(KlogKlogT) with certain restrictive assumptions. Yet, in the experiments, these algorithms seem to be performing quite poorly w.r.t. the proposed algorithm which is counter intuitive, even for 500-armed case D-TS seems to be performing better than CCB/SCB (Fig. 4(d)). Could the authors suggest an explanation for this observation? * Comparing with existing algorithms with different notions of regret for the dueling bandit problem: Are the regret definitions used in this paper directly comparable to that of some of the existing dueling bandit algorithms (e.g. RUCB, Merge-RUCB, RMED, specially those which are based on Condorcet winner assumptions)? It seems that the notion of regret used in this paper is not directly reducible to that used in these other papers with a Condorcet winner assumption. * One wonders how the regret performance gets affected if one just uses double Thompson sampling for pulling the two arms (a^(1) and a^(2) in Algorithm 1) from the whole set of K arms, instead of selecting them from the restricted set as described in line 11 and 14 of Algorithm 1. In other words, how much significant is the restrictive sampling (that is used for pulling the pair of arms) in terms of bounding the regret? * Experiments: -It might be worth comparing how the algorithm's performance scales w.r.t. other algorithms, e.g. Merge-RUCB, CCB/SCB etc. with increasing K. -It might also be worth comparing different algorithms with certain other class of pairwise preference matrices, e.g. pairwise preferences with a Borda winner, with total ordering, or preferences generated from random utility model etc. - It is not clear why ECW-RMED has higher standard deviations of regret in comparison to the proposed method since in the later case both the arms are sampled from some posterior distribution.

Confidence in this Review

2-Confident (read it all; understood it all reasonably well)


Reviewer 2

Summary

The paper proposes a Thompson Sampling (TS) style algorithm for the Copeland dueling bandit problem. The practical significance of the paper arises from the fact that, unlike the MAB setting where TS and KL-UCB have comparable performance, in the dueling bandit setting, TS-style algorithms tend to perform significantly better in practice than those using confidence bounds and this paper provides the first analysis of such an algorithm. The authors provide regret bounds for the proposed algorithm and conduct extensive experimental analysis, comparing the algorithm against every baseline published in the literature using numerous examples, carefully studying the strengths and the flaws of their algorithm. I must say I appreciate their thoroughness and honesty as far as the experimental analysis goes, which unfortunately is outside the norm for ICML/NIPS papers. The regret bound proven for the algorithm does not match the lower bound published in reference [7], but it has the enormous advantage that it is much more easily interpretable, which one could argue to be a more desirable quality than optimality, when the lower bound takes the cryptic form that it does in [7], where it is stated as the solution to some linear program. I do however have two specific issues with the proofs and I'd like the authors to address them in the rebuttal: 1- As stated, Lemma 6 is false because when applying Chernoff-Hoeffding, n (i.e. the number of samples being summed) needs to be fixed and not a random variable depending on t. It seems like the authors need a union bound on n, which means that in Equations (12-14), t^{2\alpha} should be changed to t^{2\alpha-1}. 2- In the case of Lemma 5, I do not understand how the proof can use only the asymmetric counts N^(1)_ij and not the symmetric N_ij when \theta is sampled from a posterior distribution that depends on the latter and not the former. Could you explain intuitively why you do not use the N_ij in there or are they used and I'm not seeing them for some reason?

Qualitative Assessment

The authors seem to have expended a fair bit of effort on comparing their algorithm against the existing baselines and on providing detailed arguments for their theoretical results. Moreover, the algorithm seems to perform substantially better than the baselines on most examples arising from the applications and so it is a valuable contribution from a practical point of view. The proof techniques are not really groundbreaking, but this is not a COLT submission, so I'm judging the paper more for its algorithmic contributions rather than theoretical ones.

Confidence in this Review

3-Expert (read the paper in detail, know the area, quite certain of my opinion)


Reviewer 3

Summary

The paper provides an algorithm for the dueling bandit problem with the Copeland regret definition. The authors provide an algorithm that uses a mixture of the confidence bound, and Thompson sampling methods. Crudely speaking, Thompson sampling is used either to solve sub instances that can be considered as a multi-armed bandit problem, or to break ties. The resulting regret bound is non-trivial yet is not as good as previous results. However, the algorithm seems to perform better in practice based on thorough experiments.

Qualitative Assessment

The algorithm proposed follows the general ideas of RUCB. It picks a left arm according to its optimal score, meaning it chooses an arm maximizing the optimal Copeland score, and then the right arm as the one that has potential of beating the chosen left arm. The difference from RUCB is two-fold. First, the right arm is not chosen according to a UCB type procedure but rather via a Thompson-sampling procedure. Standard techniques allow to bound the expected number of pulls for a pair of arms, leading to the regret bound. The second difference is in the selection of the left arm. Rather than choosing an arbitrary maximizer of the Copeland score, ties are broken via sampling from the posterior of the last round. The main contribution of the paper as I see it is not in presenting novel techniques but rather in a way of combining existing techniques to obtain a more (empirically) practical algorithm. The experiments provided give a convincing argument that the algorithm has a superior empirical behavior. The paper is overall well written with the possible exception of not having a clear enough high level description of the proof techniques. The above explanation needs to be extracted out and something similar should probably appear somewhere in the paper. The proofs seem to hold and are sufficiently well written, the experiments are thorough in evaluating the algorithm, and the review of the previous results is accurate and fair. Minor comment: Lemma 6 in the appendix (line 410) is not accurate. Chernoff bound can be used only when the number of trails is fixed here, the number of trials depend on the outcome of the coin tosses. However, via a union bound for all possible tries the lemma still holds but with 2\alpha-1 rather than 2\alpha. This affects only the constants in the proofs

Confidence in this Review

3-Expert (read the paper in detail, know the area, quite certain of my opinion)


Reviewer 4

Summary

Authors addressed the dueling bandit problem where the task is to pull a pair of arms in each iteration from a given set of K arms, upon which one gets to see the winner arm drawn from some underlying pairwise preferences. The objective is to identify the set of arms which are the Copeland winners or minimize regret w.r.t. the Copeland winners.

Qualitative Assessment

1. The major contribution of the work is not clear. Similar or even better regret guarantees are given previously for the same problem. For e.g. the RMED algorithm by Komiyama et al. gives a O(KlogT) regret bound which is optimal to constant factors of the regret lower bound with condorcet winner assumption where as the proposed algorithm gives a regret guarantee of O(KlogT + K^2loglogT), which is clearly theoretically worse than the performance of RMED, yet in Figure 1(a), D-TS/D-TS++ seems to be performing better than RMED which is not so convincing. Without the condorcet winner assumption, the algorithm is shown to give O(K^2logT) regret bound, where as CCB algorithm of Zoghi et al. gives O(K^2 + KlogT) regret, and SCB gives an even better regret bound of O(KlogKlogT) with certain restrictive assumptions, yet again in the experimentally these algorithms seem to be performing quite poorly w.r.t. the proposed algorithm which is counter intuitive, even for 500-armed case D-TS seems to be performing better than CCB/SCB (Fig. 4(d)), what is the reason? Even the concept of using Thompson sampling is not new, in RCS algorithm, Zoghi et al. used the similar notion to pull the first arm and the best competing arm of first arm is chosen as the second arm. 2. Comparing with existing algorithms with different notion of regret: Are the regret definition used in this paper directly comparable to that of some of the existing dueling bandit algorithms (e.g. RUCB, Merge-RUCB, RMED, specially those which are based on condorcet winner assumption), since notion of regret used in this paper is not directly reducible to that used in these other papers with condorcet winner assumption. 3. How does the regret performance get affected if one just use simple Thompson sampling for pulling the two arms (a^(1) and a^(2) in Algorithm 1) from the whole set of K arms, instead of selecting them from the restricted set as described in line 11 and 14 of Algorithm 1. In other words, how much significant is the restrictive sampling (that is used for pulling the pair of arms) in terms of bounding the regret. 4. Notation: There are few typos in the paper, for e.g. line 237: O(KlogT + K^2logT) --> O(KlogT + K^2loglogT) etc. 5. What is the practical significance of Assumption (2)? 6. Experiments: As pointed out in comment 1, some of the results/plots are not very convincing since the experimental results seem to be conflicting the theoretical guarantees. - It might be worth comparing how the algorithm's performance scales w.r.t. other algorithms, e.g. Merge-RUCB, CCB/SCB etc. with increasing K. - It might also be worth comparing different algorithms with certain other class of pairwise preference matrices, e.g. pairwise preferences with borda winner, with total ordering, or preferences generated from random utility model etc. - Its not clear why ECW-RMED has higher standard deviations of regret in comparison to the proposed method since in the later case both the arms are sampled from some posterior distribution.

Confidence in this Review

2-Confident (read it all; understood it all reasonably well)


Reviewer 5

Summary

This paper proposes two variants of Thompson Sampling for the dueling bandit problem. The authors obtain a regret bound for the two variants, and present an explanation for the advantage of one variant (D-TS+) over the other (D-TS). Experimentally, the authors show that the proposed algorithms outperform previous methods for the dueling bandit problem in some regimes.

Qualitative Assessment

In the same section, it is mentioned that theoretical analysis of TS in MAB is difficult. However, there have been some recent work in the literature which consider analysis of TS. Why double sampling used in both arms gives D-TS advantage over RCS, and significantly reduces the regret? The authors mention in Section 1 that the two selected arms in dueling bandits may be correlated, and hence to address this issue the proposed D-TS algorithm draws the two arms independently. Performance-wise, is there any advantage in the two selected arms being independent? If so, why? There are some typos/grammatical errors: in line 282, replace results with result; in line 307, replace non with not; in line 78, replace experiment with experimental. In Section 5 line 299-300, the authors mention that the arms are randomly shuffled to prevent algorithms from exploiting special structures of the preference matrix. More elaboration on this is required. One drawback of this paper is that the regret bounds for the two proposed algorithms are the same; so it is not easy to gauge the advantage of D-TS+ over D-TS, considering that D-TS is outperformed by ECW-RMED in the non-Condorcet regime. In addition, as mentioned by the authors the regret bounds are likely loose. If RUCB-based elimination seldom occurs in practice, then why use it at all? As a result of using it, both the algorithm and its regret analysis become much more complicated, and the obtained bounds are loose. Also, is it possible to obtain tighter bounds if RUCB is dropped from the algorithm? Have the authors checked the difference in performance of the proposed algorithms in numerical experiments if RUCB-elimination step is dropped? From Section 4.1, it seems that in D-TS, the second arm is obtained from comparison with the first chosen arm only. Is there any explanation for this selection? In lines 305-309: if RMED1 is optimal in Condorcet dueling bandits, then why the proposed algorithms outperform RMED1? Any intuitions into why the STD of the proposed algorithms is so much smaller than that of ECW-RMED in the non-Condorcet case? The only difference between D-TS and D-TS+ is in the tie-breaking criterion. In numerical experiments, D-TS and D-TS+ perform identically in terms of regret in the Condorcet case, but D-TS+ outperforms the other in the non-Condorcet case. How is this performance gap explained by the difference between D-TS and D-TS+?

Confidence in this Review

2-Confident (read it all; understood it all reasonably well)


Reviewer 6

Summary

This paper gives an algorithm Double-Thompson Sampling solving the dueling bandits problems. From the experimental results we can see this algorithm has a good behavior, and this is also proved in the theoretical analysis.

Qualitative Assessment

The idea of D-TS is good and the analysis result is an improvement in analyzing TS algorithms in MAB problems. The experimental results also shows that the D-TS algorithms behaves well in simulation.

Confidence in this Review

2-Confident (read it all; understood it all reasonably well)